# *Calypogeia* (Calypogeiaceae, Marchantiophyta) in Pacific Asia: Updates from Molecular Revision with Particular Attention to the Genus in North Indochina

**DOI:** 10.3390/plants11070983

**Published:** 2022-04-04

**Authors:** Vadim A. Bakalin, Yulia D. Maltseva, Frank Müller, Ksenia G. Klimova, Van Sinh Nguyen, Seung Se Choi, Aleksey V. Troitsky

**Affiliations:** 1Laboratory of Cryptogamic Biota, Botanical Garden-Institute FEB RAS, Makovskogo Street 142, 690024 Vladivostok, Russia; maltseva.yu.dm@gmail.com (Y.D.M.); ksenia.g.klimova@mail.ru (K.G.K.); 2Institut für Botanik, Technische Universität Dresden, 01062 Dresden, Germany; frank.mueller@tu-dresden.de; 3Institute of Ecology and Biological Resources, Graduate University of Science and Technology, Vietnam Academy of Science and Technology, Ha Noi 10000, Vietnam; vansinh.nguyen@iebr.ac.vn; 4Team of National Ecosystem Survey, National Institute of Ecology, Seocheon 33657, Korea; 5Belozersky Institute of Physico-Chemical Biology, Lomonosov Moscow State University, Leninskie Gory 1, 119991 Moscow, Russia

**Keywords:** Calypogeiaceae, phylogeny, taxonomy, geographic vicariants, speciation, liverworts, Hepaticae

## Abstract

*Calypogeia* is a genus in Pacific Asia that is difficult to classify taxonomically. These difficulties arise from (1) considering the presence of oil bodies as anatomical characters for taxonomic differentiation, (2) the wide occurrence of sibling, semicryptic and geographical vicariant taxa and (3) the inevitable need to organize new datasets for molecular genetic revision of the genus. The present study uses an integrative approach, including molecular genetic, morphological, chorological and ecological methods, to understand the taxonomy of the genus in Amphi-Pacific Asia. As a result, a set of new-to-science taxa was revealed, and the suite of morphological features necessary for reliable discrimination of the taxa was revised. These results are based on the study of a large set of ‘fresh’ collections suitable for molecular analysis and morphological comparison and include data on oil bodies. The most basal branch in *Calypogeia* s.l. is segregated into a new genus, Asperifolia. Descriptions of the new taxa and the key to *Calypogeia* in Vietnam are provided.

## 1. Introduction

*Calypogeia* is one of the most taxonomically complex genera in Asia, and its representatives are difficult to identify and classify [1]. These difficulties are mostly due to (1) the great taxonomic value placed on features of intracellular organelles called oil bodies for traditional (morphological) systematics (these organelles rapidly decompose, even under optimal storage conditions) and (2) the presence of a large number of semicryptic species, which differ slightly in morphology, especially in a dried state [2,3,4,5,6,7]. A review of the genus in the Sino-Himalaya based on available literature data and the study of type specimens was previously provided by our group [1]. As expected, the molecular genetic exploration undertaken in the present study revealed a number of species that need to be described as new to science, as they are not conspecific to morphologically similar taxa known from Europe and North America. In addition, the complex of taxa in the branch sister to all other *Calypogeia* should be considered the new genus *Asperfolia*. To confirm these results, we compiled a dataset of specimens for use in molecular genetic analysis. The previously obtained material from Pacific Asia that was used in the present work was only available for blue oil-bodied *Calypogeia* (mostly data from Bakalin et al. [2] and Buczkowska et al. [7], other data were scant).

The initial purpose of this work was to revise the taxonomy of the genus in Northern Indochina. However, when we added a number of *Calypogeia* specimens from adjacent regions (China, Japan and the Russian Far East) and European accessions available on GenBank to the dataset (especially species considered morphologically similar to those known in Indochina), we found that the northern part of Pacific Asia contains species that are genetically different from those described in Europe (where many have a type locality) but instead belong to other undescribed taxa. Therefore, to discriminate the taxa distributed in SE Asia from the northern allies, we first need to discriminate the Asiatic ‘northern allies’ from the morphologically similar taxa known in Europe and North America. As a result, in addition to considering the actual species found in North Indochina, it was necessary to describe a number of species from extratropical Pacific Asia, which helped to avoid confusion. In fact, a revision of the *Calypogeia* found in Pacific Asia was obtained, except for a few taxa described from Japan (which were discussed in our previous work [1], since the “fresh” material suitable for molecular genetic analysis was not available for some Japan-derived taxa). Therefore, the goal of this study was to describe species that were new to science and widespread in Pacific Asia. Additionally, we aimed to assess phytogeographic speculations on the distribution patterns of taxa based on the largest dataset for the genus using both molecular genetic and morphological methods enriched with data on the ecology and geography of the taxa.

## 2. Results

The phylogeny of *Calypogeia* taxa and closely related species was inferred from two regions of chloroplast DNA, the *trn*G intron and *trn*L−F spacer and nuclear ITS2. A list of accessions from which these sequences were available is presented in Table 1. The sets of accessions for the three DNA markers overlap only partially.

For the *trn*G intron, the alignment of 87 sequences consisted of 685 positions, among which 217 were parsimony informative, 93 were singletons and 375 were constant sites. The base frequencies across all sites were A: 0.344, C: 0.161, G: 0.129 and T: 0.365. The maximum likelihood (ML) criterion resulted in a consensus tree with a log likelihood of −3970.540. The maximum parsimony (MP) analysis yielded two equally parsimonious trees with lengths of 573 steps, a CI = 0.557809 and an RI = 0.854570. A consensus phylogenetic tree retained under Bayesian analysis (BA), along with the MP and ML bootstrap support (BS) values and the Bayesian posterior probabilities (PP) for each node are on Figure 1 (BA 50% majority rule consensus tree is on Appendix A).

All methods resulted in an almost identical tree topology. The differences concern the relative position of some low supported branches. *Calypogeia* accessions are organized into 11 supported clades; however, the relationships between these clades are poorly resolved. On the TCS haplotype network of *trn*G sequences (Figure 2), three groups of *Calypogeia* taxa were resolved.

Appendix A shows the p-distances for the *trn*G sequences as well as those for the other two markers studied between *Calypogeia* and species designated *Asperifolia*, which was previously considered to be *Calypogeia*. On the phylogenetic trees, the *Calypogeia* species clustered together and are separated by long branches from a group of other Calypogeiaceae taxa: *Mnioloma* and *Metacalypogeia*. The large gap between p-distances among the *Calypogea* species and the group of other species, including *Asperifolia*, is illustrated in Figure 3.

The *trn*L–F phylogeny is shown in Figure 4 (BA 50% majority rule tree is shown in Appendix A). The alignment of 88 *trn*L–F sequences consisted of 512 positions, among which 146 were parsimony informative, 77 were singletons and 289 were constant sites. The base frequencies across all sites were A: 0.354, C: 0.143, G: 0.189 and T: 0.344. The ML criterion resulted in a consensus tree with a log likelihood of −3059.604. The MP analysis yielded two equally parsimonious trees with lengths of 215 steps, a CI = 0.631579 and an RI = 0.894419. The arithmetic means of the log likelihoods from the Bayesian analysis for each sampling run were −3155.54 and −3159.26. The bars on the left (yellow) are based on within-*Calypogeia* distances, and those on the right are based on distances involving other genera (blue).

The ITS2 phylogenetic tree shown in Figure 5 (BA 50% majority rule tree is shown in Appendix A) was constructed from an alignment of 77 sequences that contained 553 positions, of which 149 were parsimony informative, 88 were singletons and 316 were constant sites. The base frequencies across all sites were A: 0.250, C: 0.250, G: 0.250 and T: 0.250. The ML criterion resulted in a consensus tree with a log likelihood of −3296.308. The MP analysis for ITS2 yielded three equally parsimonious trees with lengths of 455 steps, a CI = 0.712088 and an RI = 0.829870. The arithmetic means of the log likelihoods from the Bayesian analysis for each sampling run were −3360.95 and −3360.13.

The basal position on the phylogenetic tree is occupied by the species *Mesoptychia* (Jungermanniaceae), used as an outgroup, followed by *Geocalyx* (Geocalycaceae) and then *Metacalypogeia* (Calypogeiaceae).

Differences in the accessions used to produce sequences of three DNA regions do not allow the construction of a combined tree. The arrangement of *Calypogeia* taxa is not identical across the trees constructed from three DNA markers. Such discrepancies are due to short internode lengths, which resulted in low support values.

On the phylogenetic trees, the *Asperifolia taxa* are no less distant from *Calypogeia* than *Mnioloma*, another genus of Calypogeiaceae, which allows us to ascribe the rank of a genus to these liverworts.

The p-distances between *Asperifolia* and three other Calypogeiaceae genera involved in this study (*Calypogeia*, *Mnioloma* and *Metacalypogeia*), *Geocalyx* (Geocalycaceae) and *Mesoptychia* (Jungermanniaceae), as well as those between all mentioned taxa, are nearly the same (Table 2).

## 3. Discussion

### 3.1. On Calypogeia arguta Nees et Mont. and Its Relatives

The basal group of species on the phylogenetic trees separated by a long branch from other *Calypogeia* includes *C. arguta* and its relatives previously attributed to *Calypogeia* subg. *Asperifoliae* (Warnst.) R.M. Schust.

The genetic distance between this group and the rest of *Calypogeia* far exceeds the *Calypogeia* infrageneric distances (Figure 1, Figure 2, Figure 3, Figure 4,Figure 5 and Appendix A), which allows the ranking of the *Calypogeia* subg. *Asperifoliae* into the genus *Asperifolia*. The distant position of *Asperifolia arguta* and its relatives from the bulk of *Calypogeia* was also shown by morphological analysis as early as 1917 by Warnstorf [8], who proposed the unranked taxon ‘*Asperifoliae*’ within *Calypogeia*. Later, Schuster [9], following Müller [10], proposed *Calypogeia* subg. *Asperifoliae* based on that unranked taxon. Schuster (l.c.) also showed that aside from “vague and quantitative differences” in sterile plant morphology ([9]: 117), there is a valuable feature in the structure of the capsule wall outer layer, where the cells are of nearly the same width as inner cells and each fourth longitudinal wall is free of thickenings (versus epidermal cells nearly twice as wide as inner cells and every other longitudinal wall not being thickened). We were unable to check this feature in our Indochinese materials. Moreover, we did not genetically test the taxonomic position of *Calypogeia arguta* named specimens collected in East Asia; therefore, we could not identify whether ‘*C. arguta*’ from East Asia is the same taxon as that we revealed in Indochina, whether it is the same as European accessions or whether it belongs to its own species that needs to be described. Being unable to resolve this problem, we provide a description of *Asperifolia indosinica* in this paper. The species resembles *C. arguta* morphologically, although it differs in somewhat smaller cells and smaller underleaves.

### 3.2. On Calypogeia pseudosphagnicola Bakalin, Troizk. et Maltseva

The small-sized *Calypogeia* growing over *Sphagnum* in the raised oligotrophic *Sphagnum* mires (Hoohmoore) and characterized by distanced leaves was commonly automatically referred to as *C. sphagnicola* in floristic practice. However, in the southern half of the Russian Far East, the specimens resembling this species have larger midleaf cells than those of ‘true’ *C. sphagnicola* based on European accessions. Moreover, aside from *Sphagnum* mires, conditional ‘*C. sphagicola*’ in the Russian Far East was also rarely observed over moist rocky outcrops.

The molecular genetic analysis performed here showed robust differences between populations of *C. sphagnicola* and the species resembling it in the southern areas of the Russian Far East and was treated here as *C. pseudosphagnicola,* which clustered with *C. sphagnicola* f. *paludosa* (Figure 1 and Figure 4).

The latter is genetically very distant from *Calypogeia sphagnicola* f. *sphagnicola*; therefore, f. *sphagnicola* and f. *paludosa* cannot belong to the same species. Therefore, the status of *C. sphagnicola* f. *paludosa* remains unclear, and subsequent segregation is needed.

Geographically (according to available data), the areas of the two taxa somewhat overlap. However, the vast majority of localities of *C. pseudosphagnicola* lie southward of the middle part of the Khabarovsk Territory (i.e., southward of the 50-th latitude), while *C. sphagnicola* in Pacific Asia is widely distributed in NE Asia, including the Kamchatka Peninsula and Magadan Province, although it was once found in northern Sikhote-Alin (Khab-68-14-18). It is worth mentioning that although *Calypogeia sphagnicola* specimens from the Russian Far East belong to the same clade as European accessions of the same species, the genetic distance between them is quite high, and the status of Far Eastern populations of the species should be revised (as they may belong to another subspecies or even species).

### 3.3. On Calypogeia subalpina Inoue

Although *C. subalpina* was described at the species rank [11], it was not recognized at that rank even by the author of the taxon, who changed it to a subspecies of *C. neesiana* [12]. On the *trn*G and *trn*L–F trees (Figure 1 and Figure 4) as well as on the *trn*G haplotype network (Figure 2), these two species are clustered together with maximal support values; these species are absent from the ITS2 dataset. *Calypogeia subalpina* seems to be restricted by temperate and hemiboreal insular-peninsular Pacific Asia, whereas ‘true’ *C. neesiana* may be restricted in its distribution in Asia by the boreal and hemiboreal mainland (taiga biome). Some reports of *C. neesiana* from the Far East [13] may be erroneous. The typical *C. neesiana* has leaf cells elongated along the margin, whereas *C. subalpina* has longer and wider leaf margin cells in intramarginal rows. These cells sometimes appear as a rim of slightly swollen cells and resembles the rim in *C. marginella* Mitt., although the great variation in this parameter in *C. subalpina* should be noted. Another distinguishing feature of *C. subalpina* is common subrotund underleaves versus constantly transversely ellipsoidal underleaves in *C. neesiana*. The photograph from this type of specimen is provided in [1]. The specimen published in Bryophyta Selecta Exsiccata №1293 under *C. neesiana* subsp. *subalpina* (see Table 1 for the specimens examined) molecularly does not belong to the *C. subalpina-neesiana* complex and rather belongs to the *C. tosana-yoshinagana* group.

### 3.4. On the Complex Calypogeia neogaea (R.M. Schust.) Bakalin-C. kamchatica Bakalin, Troizk. et Maltseva-C. fissa (L.) Raddi

*Calypogeia fissa* subsp. *neogaea* R.M. Schust. was segregated from *C. fissa* s. str. by Schuster to place the North American populations of the taxon [9]. Later, Bakalin [14] raised its rank to the species *C. neogaea* (R.M. Schust.) Bakalin and recorded it from the Kamchatka Peninsula, where the species was estimated to be confined to very special habitats: whole-year warm (due to thermal activity of neighboring volcanoes) mesotrophic mires. After that record, the taxon was recorded several times in the Russian Far East [15], with the vast majority of localities within thermal habitats. The species status of *C. neogaea* was accepted in [16], although one year before, in the World Liverwort Checklist [17], it was treated as a subspecies. The present molecular genetic data have shown the following: (1) *C. neogaea* in North America is a taxon distinct from *C. fissa* at the species level, (2) the populations from the Russian Far East are not the same as those of *C. neogaea* and (3) the new species (mostly confined to thermal habitats in the Russian Far East) should be segregated. The latter is described here under *C. kamchatica*, which is distanced from other congeners in Figure 1, Figure 4 and Figure 5 and forms a clade with two accessions of *C. azurea*. Moreover, *Calypogeia kamchatica* is not closely related to *C. tosana* genetically despite its similar morphology. In fact, there are five taxa that are very similar morphologically: *C. fissa*, *C. kamchatica*, *C. neogaea, C. tosana* (Steph.) Steph. and *C. yoshinagana* Steph. *Calypogeia yoshinagana* is discussed below, while the morphological differences between the remaining four are minor and are mostly confined to the differences in the apical part of the leaves and underleaves. *C. tosana* commonly has shortly bifid leaves and bisbifid underleaves, while *C. fissa*, *C. neogaea* and *C. kamchatica* have acute leaf apices (rarely short bidentate, other than in *C. fissa*, which has great variation) and commonly bifid underleaves, although commonly with a blunt tooth at one or both lateral sides (underleaves are rarely short bisbifid). The most prominent differences are in the distribution of these four taxa. *Calypogeia kamchatica* is confined to the Amphi-Pacific part of the Russian Far East. Although not all records are from thermal habitats, the species is the most common (if not the only one occurring) in thermal warm mires across the Russian part of the Pacific Ring of Fire (Kamchatka plus Kuril Islands) and thermal mires in Sakhalin Island. *Calypogeia kamchatica* is formally distributed from subarctic to boreal zones, although these definitions are ‘senseless’ if the species occurs in thermal mesotrophic mires where the substrate temperature is warm all year round and mires possess intrazonal characteristics. *Calypogeia tosana* is distributed more southward, starting from the southern part of the Russian Far East (Kunashir Island and the southern flank of the Primorsky Territory) and spreads across Korea, Japan and East China to North Indochina. The species does not occur in the mesotrophic mires (which may be due to the rarity of this habitat type southward of the 40th latitude). In contrast, it is mostly confined to habitats with disturbed vegetation cover (including roadsides and steep crumbling slopes to watercourses) and is rarely observed on decaying wood, humus and litter in steep slopes in forests. *Calypogeia fissa* possesses an ecology similar to *C. tosana* and is restricted to Europe, while (the data are somewhat incomplete and may be questionable) *C. neogaea* is restricted to North America.

### 3.5. On Calypogeia tosana (Steph.) Steph. and C. yoshinagana Steph.

The closest relative of *Calypogeia tosana* is *C. yoshinagana*. The latter taxon was neglected for a long time since Hattori [18] synonymized it with *C. tosana*. This point of view was followed in the World Liverwort Checklist [17]. Dissimilar to those latter views, *C. yoshinagana* was treated as a distinct species by Bakalin et al. [1] and listed in Korea by Choi et al. [19]. *Calypogeia tosana* and *C. yoshinagana* form a well-supported clade in the phylogenetic trees and are indeed quite similar morphologically, although they may be distinguished by the leaf apex that is commonly bifid in *C. tosana* and mostly undivided in *C. yoshinagana*. In addition, *C. yoshinagana* is characterized by (1) slightly smaller cells not exceeding 40 μm in the midleaf, although midleaf cells in *C. tosana* sometimes reach 50 μm wide, and (2) a higher undivided zone in the underleaves, which commonly reach two cells high, versus one cell high in well-developed *C. tosana*. The ecology of the two species is somewhat similar, although most accessions of *C. yoshinagana* are from higher elevations (cool temperate and hemiboreal belts in the mountains), while *C. tosana*, with a few exceptions, is mostly distributed in warm-temperate to subtropical zones (and corresponding belts) and is somewhat confined to the lowlands.

### 3.6. A New Calypogeia Species from California

The study of the small liverwort collection provided by J. Shevock (CAS) and collected in California (U.S.A.) revealed small *Calypogeia* occupying a morphologically intermediate position between the pair *Calypogeia yoshinagana*-*C. tosana* and the group *Asperifolia arguta*-*A. indosinica*. The processing of the specimen for molecular analysis revealed its strong difference from both groups (Figure 1, Figure 4 and Figure 5). The new species named *Calypogeia shevockii* occupies a basal position in a clade containing most *Calypogeia* with low support on the *trn*G phylogenetic tree and a central node on the *trn*G haplotype network (Figure 2). On the *trn*L–F and ITS2 trees, its position is variable, being closest to *C. kamchatica* and *C. azurea* or *C. tosana* and *C. fissa*. The species morphologically resembles small-sized, pale-colored *C. yoshinagana*, although it has smaller, slightly more deeply divided underleaves, a commonly bifid leaf apex and pale coloration. These features may lead to its similarity with *C. tosana* and *C. fissa* (the specimen was originally identified by us as the latter taxon), as those species differ in their smaller size and dull (not translucent) plants. The range of morphological variation in the species is certainly poorly understood since only one specimen is known. Other morphologically similar species are *Asperifolia indosinica* and *A. arguta*, from which the new species differs in its smooth leaf cuticle, outer cells in the stem cross section similar to those inward (versus outer cells distinctly larger in *Asperifolia*) and smaller leaf cells. Unfortunately, the inconsistent results obtained for different genes do not permit us to describe the relationships of *C. shevockii* more definitely.

### 3.7. On Calypogeia japonica Steph.

On the *trn*G tree, *C. japonica* is included in a basal clade containing six other species but separated from them with maximum support by a long branch (Figure 1). This species occupied a basal position to all other *Calypogeia* on the *trn*L–F tree (Figure 4). *Calypogeia japonica* is very distinct from other *Calypogeia* due to biconcentric oil bodies that are not known in other taxa of the genus. In the absence of oil bodies, the species may be mistaken for a morphologically similar species, *C. muelleriana*, due to its rounded to obtuse leaf apices and distinctly divided underleaves (although far less deeply as in *C. tosana* and its morphological relatives). *Calypogeia muelleriana* is widely distributed in the North Holarctic (one of the most common taxa in North Europe), although it is quite rare in NE Asia and does not appear to occur southward of the 50th latitude in hemiboreal and temperate Pacific Asia. In contrast, *C. japonica* occurs southward of the 45–47th latitudes in the insular part (South Kurils plus Japan) and from the Korean Peninsula in the continental mainland to North Indochina. However, presumably, we estimate that the area of *C. japonica* spreads northwards by the islands in the North Pacific until North Kurils and even probably to the Commander Archipelago, where the study of fresh material is strongly required to properly identify the specimens. Other features are minor and qualitative: *C. japonica* has more or less translucent shoots and wavy leaves in comparison to the opaque and more rigid shoots of *C. muelleriana*.

### 3.8. On Calypogeia pseudocuspidata Bakalin, Frank Müll. et Troizk.

This species occupies a sister position to the abovementioned clade with or without the *C. subalpina*-*C. neesiana* complex on *trn*G and *trn*L–F trees, although it has quite a different position among *Calypogeia vietnamica* and *C. granulata* on the ITS2 tree (Figure 1, Figure 4 and Figure 5). *Calypogeia pseudocuspidata* is characterized by distinct morphology. It somewhat resembles *C. cuspidata* with its relatively narrow, shortly decurrent underleaves, acute to shortly bicuspidate leaf apices (although never rounded, as sometimes occurs in *C. cuspidata*) and the upper third of leaves turned to the ventral side of the shoot. The two species are clearly different due to larger, somewhat swollen cells along leaf (also in the underleaf, although to a lesser extent) margins, a rigid texture and a brownish color in the fresh state. Swollen cells along the leaf margin are a rare feature in the *Calypogeia* genus but are well pronounced in *C. pseudocuspidata* and *C. marginella*. To a lesser extent, the larger cells along the leaf margin can be observed in a suite of taxa, including *C. subalpina* and *C. pseudointegristipula*. The distribution of the taxon may be underestimated since, aside from Vietnam, it was found in the *Calypogeia* collection from Myanmar and may be distributed in other areas of the Sino-Himalayas.

### 3.9. On the Distribution of Calypogeia integristipula Steph. in the Pacific

The most unexpected record was that of the specimens named *Calypogeia integristipula* from the southern part of the Russian Far East, which actually belong to a previously undescribed species referred to here as *C. pseudointegristipula*. Therefore, the identity of specimens named *C. integristipula* from Pacific Asia should be questioned. Indeed, *C. integristipula* was described from Europe and Japan based on several specimens as indefinite syntypes [20]. Later, the taxon was lectotypified by a specimen from Germany [21], and the lectotypifacation was followed by Grolle [22]. Indeed, there are some specimens from the Russian Far East that are genetically similar to the accessions from Europe. However, these specimens arise from the continental mainland of the Far East. Two *C. pseudointegristipula* specimens from Sakhalin and South Kurils on *trn*G and *trn*L–F trees were placed in the same clade as “true” *C. integristipula*. On the ITS2 tree, the position of these species is unclear.

Morphologically, *C. pseudointegristipula* is similar to *C. integristipula* but may be distinguished by its thickened cell walls in the marginal cell rows in the leaf, distinct trigones in the leaf cells and distinctly emarginate to shortly divided underleaves. Since all specimens collected from South Kurils and the southern part of the Sakhalin Island belong to *C. pseudointegristipula,* we may expect its occurrence in Japan and Korea, where this species presumably may substitute for *C. integristipula* (the latter then remains a circumboreal taxon that does not spread to the East Asian floristic region).

### 3.10. On Blue Oil-Bodied Calypogeia

Deep blue oil bodies are known in several species of *Calypogeia* [2,7] and are easily observable when the material is fresh. The blue color of oil bodies is evident even in the field due to distinctly bluish coloration in the shoot apices. The coloration of oil bodies is due to azulene derivate production [23] and might suggest the evolutionary value of this feature and that all blue oil-bodied *Calypogeia* tentatively form a peculiar monophyletic group within the genus. However, it was found that blue oil-bodied *Calypogeia* are scattered across all phylogenetic trees; therefore, this feature might appear several times, or in contrast, the production of azulene might be suppressed in several species. Indeed, the blue oil bodies are not only found in *Asperifolia* and the *C. pseudocuspidata*-*C.* and *integristipula*-*C. melleriana* group and their closest relatives. To visualize their distribution across the phylogenetic tree, *Calypogeia* species possessing blue oil bodies are marked with asterisks in the *trn*G phylogenetic tree (Figure 1).

### 3.11. On the Genus Calypogeia in Vietnam

The last list of liverworts in Vietnam [24] contains information on eight species of *Calypogeia* recorded in the country in the literature (including our own papers [7] and [2,25]): *C. aeruginosa* Mitt., *C. arguta* Nees & Mont., *C. azurea* Stotler & Crotz, *C. granulata* Inoue, *C. japonica* Steph., *C. sinensis* Bakalin et Buczkovska, *C. tosana* (Steph.) Steph. and *C. vietnamica* Bakalin et Vilnet. This list should be updated as follows.

The reports of *Calypogeia arguta*, as shown in the present account, should be transferred to *Asperifolia indosinica*. True *A. arguta* appears to be restricted to Europe and does not occur in Asia. The first report of *C. azurea* in Vietnam was provided by Shu et al. [26]. The authors wrote that “*Calypogeia azurea* is immediately separated from the known species of *Calypogeia* in Vietnam by the blue oil bodies and bilobed underleaves.” (l.c.: 414). Since that publication, the species was found not to occur in Amphi-Pacific Asia, while several other species with blue oil bodies are distributed in East Asia [7]. True *C. azurea* seems to be restricted to Europe (probably slightly entering Siberia by dark coniferous forests). Currently, there are three species with blue oil bodies in the Vietnam flora, while the report of *C. azurea* in Shu et al. [26] presumably belongs to *C. sinensis*.

Three more species should be added to the list of *Calypogeia* known in Vietnam: *C. apiculata*, *C. pseudocuspidata* and *C. cuspidata*. The description of *C. pseudocuspidata* is provided in the present paper. The morphology of *C. apiculata* is described and illustrated in Bakalin et al. [1]. The plants we named *C. apiculata* in North Vietnam may not be identical to the ‘true’ *C. apiculata* described from Java Island. We were not able to test the populations from Java genetically. Moreover, the populations from Vietnam and China (the Chinese accession citations are in Table 1) may be identical to plants with the invalid name ‘*C. gollanii*’ Steph. ex Bonner, as discussed in [1]. The occurrence of *C. cuspidata* is even more problematic in Sino-Himalaya than that of *C. apiculata*. This taxon was described by Hawaii [27] and listed under *Kantius cuspidatus* Steph. It is questionable whether this species may have occurred in Sino-Himalaya. Moreover, there are two more names (*C. confertifolia* Steph. and *C. hawaica* Steph.) with similar descriptions from Hawaii that may be conspecific with *C. cuspidata* (for further discussion, see [1]). *Calypogeia cuspidata* is very similar morphologically to *C. apiculata,* which differs in its smooth cuticle (versus finely verruculose). Other differences are minor and unstable and include no or barely (versus up to 1/3 of stem width) decurrent underleaves, more densely and subimbricate (versus contiguous to somewhat distant) inserted leaves and wider (1.4 as wide as a stem versus 1.1–1.2 as wide as a stem) underleaves.

Noticeably, all reported *Calypogeia* taxa were found in the northern part of the country (the genus itself was first reported in Vietnam in 2017 by Shu et al. [26], based on two species records). *Calypogeia* likely occurs in the southern part of the country, but there are still no data to support this hypothesis.

Therefore, ten species are now known in Vietnam, and all belong to East Asian or SE Asian floral elements. Earlier [1], we provided a key to identifying all species of *Calypogeia* that were known and expected in Sino-Himalaya and the surrounding areas (including North Vietnam). That key certainly did not include taxa described in the present work. Moreover, it is quite complicated to use keys containing many species, with many couplets and many morphologically similar geographic vicariants. Thus, we provide an updated key to *Calypogeia* and *Asperifolia* species known in Vietnam.

#### 3.11.1. Key to *Calypogeia* and *Asperifolia taxa* in Vietnam

1. Underleaves as large as leaves or larger, transversely ellipsoidal, with an apex entire to shallowly emarginate, oil bodies deep blue to purple and purple brown … *C. aeruginosa.*

1. Underleaves commonly smaller than leaves (rarely as large as leaf) and distinctly 2–4-lobed, oil bodies deep blue to colorless and brownish, never purple … 2.

2. Leaves with rounded to obtuse apex, oil bodies granulate with central eye (biconcentric) … *C. japonica.*

2. Leaves with acute, apiculate to bifid apex, oil bodies granulate to botryoidal, never with central eye … 3.

3. Leaf cuticle papillose to verruculose … 4.

3. Leaf cuticle smooth … 5.

4. Leaves with constantly bicuspidate apex, outer stem cells in the cross section distinctly larger than inner cells, underleaves deeply bisbifid … *A. indosinica.*

4. Leaves with acute (very rarely bicuspidate) apex, outer stem cells in the cross section as large as inner cells, underleaves bifid … *C. apiculata.*

5. Underleaves as large as leaves or smaller to 2/3 of leaf size, undivided zone in underleaf over 5–7 cells high … *C. vietnamica.*

5. Underleaves much smaller than leaves (never exceeding ¼ of leaf size) … 6.

6. Cells along the leaf margin commonly distinctly larger than those inward, thick-walled plants commonly brownish and brownish greenish when fresh … *C. pseudocuspidata.*

6. Cells along the leaf margin as large as cells inward, not swollen, with merely thin walls, plants pale greenish to greenish yellowish and bluish when fresh … 7.

7. Underleaves bifid, leaves acute, oil bodies granulate, colorless … *C. cuspidata.*

7. Underleaves bisbifid (rarely bifid in smaller plants), leaves bicuspidate to acute, oil bodies botryoidal (colorless) to coarsely granulate (deep blue in color) and very finely granulate (then commonly brownish) … 8.

8. Oil bodies very finely granulate, brownish to bluish brownish … *C. granulata.*

8. Oil bodies coarsely granulate to botryoidal, colorless to deep blue … 9.

9. Oil bodies botryoidal, colorless, leaves almost constantly bicuspidate, underleaves deeply bisbifid … *C. tosana.*

9. Oil bodies coarsely granulate, deep blue, leaf apex acute to bicuspidate, underleaves deeply to more shortly bisbifid … *C. sinensis.*

#### 3.11.2. Annotated List of *Calypogeia* and *Asperifolia taxa* in Vietnam

Below, we provide a list of *Calypogeia* and *Asperifolia* species known in Vietnam that have specimens examined and corresponding references to recent descriptions and illustrations. The specimens examined are limited to one for each locality where the species was recorded. The newly described species are listed but not annotated.

***Asperifolia indosinica*** Troizk. et Bakalin, the present paper.

***Calypogeia aeruginosa*** Mitt., J. Proc. Linn. Soc., Bot. 5 (18): 107, 1860 [1861].

Basionym: *Kantius aeruginosus* (Mitt.) Steph., Hedwigia 34: 55, 1895.

Type: India, Sikkim, 12,000 ped. alt. (4000 m a.s.l.), J.D. Hooker, no. 1319 (isotype: G [G00064244/5286!]).

Descriptions in [28].

Illustrations in [28] (Figures 1 and 2), [1] (Figure 2A); the present paper, Figure 6.

Specimens examined: Vietnam, Lao Cai Province, Hoang Lien Range, Hoang Lien National Park, Phan Xi Pan Peak Area (22°18′15″ N 103°46′30″ E), 3050 m a.s.l., 20 April 2017, V.A. Bakalin & K.G. Klimova V-8-36-17 (VBGI), main path to the Phan Xi Pan Peak (22°18′46″ N 103°45′51″ E), 2860 m a.s.l., 22 April 2017, V.A. Bakalin & K.G. Klimova V-11-12-17 (VBGI), (22°18′28″ N 103°46′25″ E), 2846 m a.s.l., 3 April 2018, V.A. Bakalin & K.G. Klimova V-17-14-18 (VBGI).

Ecology: Open to partly shaded moist cliffs, including those near streams in oro-subtropical forests with some warm-temperate floral elements with rocky outcrops.

***Calypogeia apiculata*** (Steph.) Steph., Bull. Herb. Boissier (sér. 2) 8 (9): 668 (400), 1908.

Basionym: *Kantius apiculatus* Steph., Hedwigia 34 (2): 51, 1895.

Type: Java, Prof. Stahl (lectotype (designated in [1]): G [G00061103!]).

Descriptions in [1].

Illustrations in [1] (Figures 2D,E and 3H,I); the present paper, Figure 7.

Specimens examined: Vietnam, Lao Cai Province, Hoang Lien Son Range (22°25′14″ N 103°46′46″ E), 1325 m a.s.l., 16 April 2017, V.A. Bakalin & K.G. Klimova V-1-34-17 (VBGI), (22°23′54″ N 103°47′07″ E), 1410 m a.s.l., 16 April 2017, V.A. Bakalin & K.G. Klimova V-2-11-17 (VBGI), Hoang Lien National Park (22°21′20″ N 103°47′33″ E), 1520 m a.s.l., 17 April 2017, V.A. Bakalin & K.G. Klimova V-3-14-17 (VBGI), one of the ways to the Phan Xi Pan Peak (22°19′35″ N 103°46′57″ E), 2210 m a.s.l., 19 April 2017, V.A. Bakalin & K.G. Klimova V-5-71-17 (VBGI), Hoang Lien National Park (22°20′58″ N 103°45′49″ E), 1840 m a.s.l., 21 April 2017, V.A. Bakalin & K.G. Klimova V-10-28-17 (VBGI), southern slope of Phan Xi Pan Peak (22°18′32″ N 103°46′44″ E), 2670 m a.s.l., 4 April 2018, V.A. Bakalin & K.G. Klimova V-19-27-18 (VBGI); Vietnam, Lai Châu Province, Pu Ta Leng Mt. Range (22°25′33″ N 103°34′16″ E), 1703 m a.s.l., 26 March 2018, V.A. Bakalin & K.G. Klimova V-3-9-18 (VBGI), (22°24′56″ N 103°35′47″ E), 2347 m a.s.l., 27 March 2018, V.A. Bakalin & K.G. Klimova V-7-8-18 (VBGI), Banh Hang Stream valley (22°30′09.3″ N 103°33′15.6″ E), 1356 m a.s.l., 4 April 2019, V.A. Bakalin & K.G. Klimova V-2-5-19 (VBGI), (22°30′09.8″ N 103°34′36.0″ E), 2200-2300 m a.s.l., 5 April 2019, V.A. Bakalin & K.G. Klimova V-3-4-19 (VBGI), (22°29′53.5″ N 103°33′55.0″ E), 1582 m a.s.l., 8 April 2019, V.A. Bakalin & K.G. Klimova V-15-5-19 (VBGI).

Ecology: Open to partly shaded moist cliffs and boulders, including those near streams in oro-subtropical forests, rarely scattered *Abies delavayi* stands with *Rhododendron* trees and *Sinobambusa* thickets in the understorey.

***Calypogeia cuspidata*** (Steph.) Steph., Bull. Herb. Boissier (sér. 2) 8 (9): 669 (401), 1908.

Basionym: *Kantius cuspidatus* Steph., Bull. Herb. Boissier 5 (10): 846, 1897.

Type: Hawaii, Heller 2308 (lectotype (designated in [1]): G [G00069713!].

Descriptions in [1].

Illustrations in [1] (Figures 6N–Q, 7A–F and 9G–M); the present paper, Figure 8.

Specimens examined: Vietnam, Cao Bằng Province, Phia Oac–Phia Den National Park (22°36.818′N 105°52.148′E), 1772 m a.s.l., 26 March 2020, V.A. Bakalin & K.G. Klimova, V-23-13-20 (VBGI).

Ecology: Open moist cliffs in oro-subtropical forests with dense bamboo understorey.

***Calypogeia granulata*** Inoue, J. Jap. Bot. 43 (10/11): 468, 1968.

Type: Japan, Saitama Prefecture, Kuroyama, 500 m a.s.l., 24 June 1968, H. Inoue, 18004 (holotype: TNS [174361!]; isotype: G [G00114896!]).

Descriptions in [1,29].

Illustrations in [29] (Figure 1), [1] (Figures 4A–K and 5A–E); the present paper, Figure 9.

Specimens examined: Vietnam, Lao Cai Province, Hoang Lien Range (22°23′54″ N 103°47′07″ E), 1410 m a.s.l., 16 April 2017, V.A. Bakalin & K.G. Klimova V-2-10-17 (VBGI), Hoang Lien Range, Hoang Lien National Park (22°21′20″ N 103°47′33″ E), 1520 m a.s.l., 17 April 2017, V.A. Bakalin & K.G. Klimova V-3-83-17 (VBGI), (22°20′55″ N 103°46′03″ E), 1700–1900 m a.s.l., 15 March 2016, V.A. Bakalin V-1-16-16 (VBGI), one of the ways to the Phan Xi Pan Peak (22°19′10″ N 103°47′17″ E), 2030 m a.s.l., 18 April 2017, V.A. Bakalin & K.G. Klimova V-4-1-17 (VBGI), (22°19′35″ N 103°46′57″ E), 2210 m a.s.l., 19 April 2017, V.A. Bakalin & K.G. Klimova V-5-59-17 (VBGI), main path to the Phan Xi Pan Peak (22°20′10″ N 103°46′46″ E), 2060 m a.s.l., 19 April 2017, V.A. Bakalin & K.G. Klimova V-6-6-17 (VBGI), Phan Xi Pan Peak Area (22°18′15″ N 103°46′30″ E), 3050 m a.s.l., 20 April 2017, V.A. Bakalin & K.G. Klimova V-8-67-17 (VBGI), Hoang Lien National Park (22°20′58″ N 103°45′49″ E), 1840 m a.s.l., 21 April 2017, V.A. Bakalin & K.G. Klimova V-10-13-17 (VBGI), main path to the Phan Xi Pan Peak (22°19′12″ N 103°46′11″ E), 2610 m a.s.l., 22 April 2017, V.A. Bakalin & K.G. Klimova V-12-5-17 (VBGI), Lai Châu Province, Pu Ta Leng Mt. Range (22°24′56″ N 103°35′47″ E), 2347 m a.s.l., 27 March 2018, V.A. Bakalin & K.G. Klimova V-7-52-18 (VBGI), Banh Hang Stream valley (22°30′09.8″ N 103°34′36.0″ E), 2200–2300 m a.s.l., 5 April 2019, V.A. Bakalin & K.G. Klimova V-3-4-19 (VBGI), S-facing slope of Bach Moc Luong Tu Mt. (22°30′10.0″ N 103°35′06.5″ E), 2800–2900 m a.s.l., 6 April 2019, V.A. Bakalin & K.G. Klimova V-8-4-19 (VBGI), Banh Hang Stream valley (22°30′02.7″ N 103°34′18.7″ E), 2022 m a.s.l., 7 April 2019, V.A. Bakalin & K.G. Klimova V-13-8-19 (VBGI), Sơn La Province, Xuan Nha Nature Reserve, area of Long Sap Army Gate, NW-facing slope of Dinh Pha Luong Mt. (20°40′28.9″ N 104°37′53.6″ E), 1700–1800 m a.s.l., 11 April 2019, V.A. Bakalin & K.G. Klimova V-21-16-19 (VBGI), area of Long Sap Army Gate (20°40′53.4″ N 104°38′18.9″ E), 1500 m a.s.l., 12 April 2019, V.A. Bakalin & K.G. Klimova V-24-17-19 (VBGI), Hà Giang Province, Tay Con Linh Range, Tay Con Linh Nature Reserve (22°47′47.7″ N 104°48′53.4″ E), 1990 m a.s.l., 21 March 2020, V.A. Bakalin & K.G. Klimova V-9-16-20 (VBGI), (22°47′50.3″ N 104°48′40.9″ E), 2191 m a.s.l., 21 March 2020, V.A. Bakalin & K.G. Klimova V-11-3-20 (VBGI), area near Tay Con Linh Mt. top (22°48′06.7″ N 104°48′22.7″ E), 2423 m a.s.l., 22 March 2020, V.A. Bakalin & K.G. Klimova V-14-15-20 (VBGI), Cao Bằng Province, Phia Oac–Phia Den National Park (22°36.502′N 105°52.270′E), 1573 m a.s.l., 25 March 2020, V.A. Bakalin & K.G. Klimova V-21-15-20 (VBGI), Lạng Sơn Province, Mau Son mountain area, Phia Po Mt. top (21°50′10.2″ N 106°57′39.6″ E), 1537 m a.s.l., 31 March 2020, V.A. Bakalin & K.G. Klimova V-31-2-20 (VBGI).

Ecology: Open to partly shaded mesic to moist cliffs and their crevices, fine soil along streams and trails, living and decaying tree trunks in oro-subtropical forests, commonly with a dense bamboo understorey.

***Calypogeia japonica*** Steph., Sp. Hepat. (Stephani) 6: 448, 1924.

Type: Japan, “Japonia, Uematsu” (neotypified by Furuki and Ota [30]: G [G00047413/9720!]).

Descriptions in [1,31].

Illustrations in [30] (Figures 1–10), [31] (Figures 1 and 2) and [1] (Figures 6R–Z, AA–AF, 7J–L).

Specimens examined: Vietnam, Lao Cai Province, Hoang Lien Son Range (22°25′14″ N 103°46′46″ E), 1325 m a.s.l., evergreen south subtropical mountain forest in stream valley, open moist cliff near stream, 16 April 2017, V.A. Bakalin & K.G. Klimova, V-1-31-17 (VBGI).

***Calypogeia pseudocuspidata*** Bakalin, Frank Müll. et Troizk., the present paper.

***Calypogeia sinensis*** Bakalin & Buczk. PLoS ONE 13(10): e0204561 [7], 2018.

Type: China, Guizhou Province, Duyun Municipality (26°22.383′N, 107°21.35′E), 1300 m a.s.l., 22 November 2013, V.A. Bakalin, China-56-77-13 (holotype: VBGI!; isotype: POZW!).

Descriptions in [7].

Illustrations in [7] (Figure 7); the present paper, Figure 10.

Specimens examined: Vietnam, Lao Cai Province, Hoang Lien Range, Hoang Lien National Park, one of the ways to the Phan Xi Pan Peak (22°19′35″ N 103°46′57″ E), 2210 m a.s.l., 19 April 2017, V.A. Bakalin & K.G. Klimova V-5-24-17 (VBGI), Lai Châu Province, Pu Ta Leng Mt. summit (22°25′22″ N 103°36′14″ E), 3050 m a.s.l., 30 March 2018, V.A. Bakalin & K.G. Klimova V-11-25-18 (VBGI), Pu Ta Leng Mt. Range (22°24′28″ N 103°36′37″ E), 2394 m a.s.l., 30 March 2018, V.A. Bakalin & K.G. Klimova V-13-3-18 (VBGI), Lao Cai Province, Hoang Lien Range, Hoang Lien National Park, area near Phan Xi Pan Peak (22°18′13″ N 103°46′32″ E), 3105 m a.s.l., 3 April 2018, V.A. Bakalin & K.G. Klimova V-18-17-18 (VBGI), Hoang Lien National Park (22°20′22.6″ N 103°46′40.2″ E), 2014 m a.s.l., 5 April 2018, V.A. Bakalin & K.G. Klimova V-21-23-18 (VBGI), Hanoi Capital Province, Ba Vi National Park (21°03′39″ N 105°21′47″ E), 1072 m a.s.l., 8 April 2018, V.A. Bakalin V-22-7-18 (VBGI), Sơn La Province, Xuan Nha Nature Reserve, area of Long Sap Army Gate, NW-facing slope of Dinh Pha Luong Mt. (20°40′28.9″ N 104°37′53.6″ E), 1700–1800 m a.s.l., 11 April 2019, V.A. Bakalin & K.G. Klimova V-21-26-19 (VBGI), Xuan Nha Nature Reserve, area of Long Sap Army Gate (20°41′11.5″ N 104°37′25.8″ E), 1200–1300 m a.s.l., 12 April 2019, V.A. Bakalin & K.G. KlimovaV-23-15-19 (VBGI), (20°40′53.4″ N 104°38′18.9″ E), 1500 m a.s.l., 12 April 2019, V.A. Bakalin & K.G. Klimova V-24-12-19 (VBGI), Hà Giang Province, Tay Con Linh Range, Tay Con Linh Nature Reserve (22°47′43.0″ N 104°49′16.2″ E), 1672 m a.s.l., 20 March 2020, V.A. Bakalin & K.G. Klimova V-5-7-20 (VBGI), (22°47′47.7″ N 104°48′53.4″ E), 1990 m a.s.l., 21 March 2020, V.A. Bakalin & K.G. Klimova V-9-20-20 (VBGI), (22°47′50.3″ N 104°48′40.9″ E), 2191 m a.s.l., 21 March 2020, V.A. Bakalin & K.G. Klimova V-11-7-20 (VBGI), Cao Bằng Province, Phia Oac–Phia Den National Park (22°36.502′N 105°52.270′E), 1573 m a.s.l., 25 March 2020, V.A. Bakalin & K.G. Klimova V-21-12-20, V-21-26-20 (VBGI), (22°36.443′N 105°52.039′E), 1691 m a.s.l., 26 March 2020, V.A. Bakalin & K.G. Klimova V-24-7-20 (VBGI), (22°36′52.6″ N 105°52′05.3″ E), 1796 m a.s.l., 27 March 2020, V.A. Bakalin & K.G. Klimova V-26-5-20, V-26-17a-20 (VBGI), (22°36′30.2″ N 105°52′08.4″ E), 1593 m a.s.l., 28 March 2020, V.A. Bakalin & K.G. Klimova V-27-4-20 (VBGI).

Ecology: Open to partly shaded moist boulders and cliffs (also covered with fine soil), including those near streams, moist trunks of living and decaying tree trunks, rarely clayish soil on steep slopes, in oro-subtropical and, rarely, tropical forests.

***Calypogeia tosana*** (Steph.) Steph., Bull. Herb. Boissier (sér. 2) 8 (9): 678 (410), 1908.

Type: Japan, Tosa Makino (LECTOTYPE (designated here): G [G00047274/26013, packet b!]).

Descriptions in [1].

Illustrations in [32] (Figure 18) and [1] (Figures 4AD–AH and 8F–I); the present paper, Figure 11.

Specimens examined: Vietnam, Lao Cai Province, Hoang Lien Range, Hoang Lien National Park, (22°20′55″ N 103°46′03″ E), 1700–1900 m a.s.l., 15 March 2016, V.A. Bakalin V-1-16-16 (VBGI), (22°21′20″ N 103°47′33″ E), 1520 m a.s.l., 17 April 2017, V.A. Bakalin & K.G. Klimova V-3-73-17 (VBGI), Hoang Lien National Park, Phan Xi Pan Peak Area (22°18′15″ N 103°46′30″ E), 3050 m a.s.l., 20 April 2017, V.A. Bakalin & K.G. Klimova V-8-64-17 (VBGI), southern slope of Phan Xi Pan Peak (22°18′32″ N 103°46′44″ E), 2670 m a.s.l., 4 April 2018, V.A. Bakalin & K.G. Klimova V-19-11-18 (VBGI), Hanoi Capital Province, Ba Vi National Park (21°03′39″ N 105°21′47″ E), 1072 m a.s.l., 8 April 2018, V.A. Bakalin V-22-8-18 (VBGI), Lai Châu Province, Banh Hang Stream valley (22°30′00.38″ N 103°33′29.98″ E), 1509 m a.s.l., 8 April 2019, V.A. Bakalin & K.G. Klimova V-16-2-19 (VBGI), Cao Bằng Province, Phia Oac–Phia Den National Park (22°36.818′N 105°52.148′E), 1772 m a.s.l., 26 March 2020, V.A. Bakalin & K.G. Klimova V-23-2-20, V-23-16-20 (VBGI).

Ecology: Open moist cliffs near watercourses to (more frequently) clayish soil in steep slopes, stream banks and trails in oro-subtropical forests.

***Calypogeia vietnamica*** Bakalin et Vilnet Herzogia 32 (1): 225, 2019.

Type: Vietnam, Lao Cai Province, SaPa District, San Sa Ho Commune, Hoang Lien Range, Phan Xi Pang Peak area (22°18.45′N, 103°46.567′E), 2900 m a.s.l., 20 April 2017, V.A. Bakalin & K.G. Klimova, V-9-23-17 (holotype: VBGI!).

Descriptions in [2].

Illustrations in [1] (Figures 3 and 4).

Specimens examined: Vietnam, Lao Cai Province, Hoang Lien Son Range, Hoang Lien National Park, Phan Xi Pan Peak Area (22°18′15″ N 103°46′30″ E), 3050 m a.s.l., 20 April 2017, V.A. Bakalin & K.G. Klimova V-8-48-17 (VBGI), (22°18′27″ N 103°46′34″ E), 2900 m a.s.l., 20 April 2017, V.A. Bakalin & K.G. Klimova V-9-23-17 (VBGI), Lai Châu Province, Pu Ta Leng Mt. summit (22°25′22″ N 103°36′14″ E), 3050 m a.s.l., 30 March 2018, V.A. Bakalin & K.G. Klimova V-11-45-18 (VBGI), the ridge line (22°25′18″ N 103°36′26″ E), 2930 m a.s.l., 30 March 2018, V.A. Bakalin & K.G. Klimova V-12-10-18, Pu Ta Leng Mt. Range (22°24′28″ N 103°36′37″ E), 2394 m a.s.l., 30 March 2018, V.A. Bakalin & K.G. Klimova V-13-3-18 (VBGI), Lao Cai Province, Hoang Lien Son Range, Hoang Lien National Park, area near Phan Xi Pan Peak (22°18′13″ N 103°46′32″ E), 3105 m a.s.l., 3 April 2018, V.A. Bakalin & K.G. Klimova V-18-18-18 (VBGI).

Ecology: Open to partly shaded moist cliffs, including those near streams, moist humus on steep slopes, moist trunk bases in part shade in oro-subtropical forests and dense communities composed of *Sinobambusa* and *Rhododendron*.

### 3.12. On the Genus Calypogeia in Myanmar

The liverwort flora of Myanmar is poorly known. To date, no member of *Calypogeia* has been reported from Myanmar.

In the course of biodiversity transect studies along elevational gradients in Myanmar from 2012–2014 (Mt Victoria, Natma Taung National Park, Chin State; Hponyin Razi and Hponkan Razi NW of Putao, Kachin State), Georg Miehe and his colleagues collected an extensive number of bryophyte specimens. This material was sent to F. Müller for further processing and identification.

Among the specimens that were found, at least five species of *Calypogeia* s.l. in the collections were not ‘fresh’, thus the study of the oil bodies was not possible. Only in one specimen (J. Kluge & P.K. Kine 14-026-043a-H) were remnants of oil bodies observed, and the blue color was distinctive. All recorded *Calypogeia* specimens were gathered in upper elevations of the mountains at 2000–3588 m.

The following species are represented in the material and all of them represent first records for Myanmar: *Asperifolia indosinica* Bakalin et Troizk., *Calypogeia lunata* Mitt., *Calypogeia pseudocuspidata* Bakalin, Frank Müll. et Troizk., *Calypogeia tosana* (Steph.) Steph., and *Calypogeia vietnamica* Bakalin et Vilnet.

Specimens examined:

*Asperifolia indosinica* Bakalin et Troizk.: Myanmar, Hponyin Razi, (27°36′25.2″ N 96°59′03.7″ E) 2039 m a.s.l., *Lithocarpus-Magnolia* forest, 12 November 2013, G. Miehe et al. 13-091-144 (DR).

*Calypogeia lunata* Mitt.: Myanmar, Hponyin Razi (27°37′34.8″ N 96°58′40.1″ E) 3016 m a.s.l., upper montane *Abies-Rhododendron* forest, 25 October 2013, G. Miehe et al. 13-070-044-B (DR), (27°37′32.4″ N 96°58′40.4″ E) 3010 m a.s.l., upper montane *Abies-Rhododendron* forest, 25 October 2013, G. Miehe et al. 13-071-043-B (DR), (27°36′51.5″ N 96°58′55.4″ E) 2433 m a.s.l., *Magnolia-Fagaceae*-bamboo forest, 03 November 2013, G. Miehe et al. 13-083-074-D (DR), (27°36′46.6″ N 96°58′54.5″ E) 2371 m a.s.l., *Magnolia-Fagaceae*-bamboo forest, 02 November 2013, G. Miehe et al. 13-081-094-G (DR, VBGI), mixed with *C. pseudocuspidata*.

*Calypogeia pseudocuspidata* Bakalin, Frank Müll. et Troizk., see the information in the description of the species in the present paper.

*Calypogeia tosana* (Steph.) Steph.: Myanmar, Hponyin Razi (27°36′22.3″ N 96°59′04.7″ E) 2039 m a.s.l., *Lithocarpus-Magnolia* forest, 10 November 2013, G. Miehe et al. 13-089-142-B (DR).

*Calypogeia vietnamica* Bakalin et Vilnet: Myanmar, Kachin State, Hponkan Razi (27°30′21.3″ N 96°56′06.8″ E) 3588 m a.s.l., evergreen broadleaved forest, 11 November 2014, J. Kluge & P.K. Kine 14-026-043a-H (DR).

## 4. Taxonomic Treatment

The treatment includes two groups of taxa: (1) taxa newly described in the present paper and (2) taxa whose classification should be clarified in light of newly obtained data.

***Asperifolia*** (Warnst.) Troizk., Bakalin, Maltseva, ***comb. nov.***

Basionym: *Calypogeia* [unranked] *Asperifoliae* Warnst., Bryol. Z. 1 (7): 111, 1917 (=*Calypogeia* subg. *Asperifolia* (Warnst.) R.M. Schust., Hepat. Anthocerotae N. Amer. 2: 115, 1969)

Type of species: ***Asperifolia arguta*** (Nees et Mont.) Troizk., Bakalin et Maltseva, ***comb. nov.***

Basionym: *Calypogeia arguta* Nees et Mont., Naturgesch. Eur. Leberm. 3: 24, 1838.

Other taxa: *Asperifolia indosinica* (see below); ***Asperifolia sullivantii*** (Austin) Troizk., Bakalin et Maltseva, ***comb. nov.*** Basionym: *Calypogeia sullivantii* Austin, Hepat. bor.-amer.: 19, 1873.

***Asperifolia indosinica*** Bakalin et Troizk. ***sp. nov.***

Description. Plants are soft, greenish to whitish greenish, greatly varying in size, in loose patches, creeping, loosely attached to the substratum and 10–20 mm long and 0.8–0.4 mm wide. Rhizoids virtually absent or very few. Stem cross section transversely ellipsoidal, 100–120 μm high and 150–170 μm wide, outer cells larger than inner, 22–35 μm in diameter, in the dorsal side with thicker walls, inner cells 20–30 μm in diameter, with slightly thickened walls, trigones small concave. Leaves subhorizontally inserted, contiguous to somewhat distant, nearly plane, not decurrent dorsally, shortly, but clearly decurrent ventrally, when flattened in the slide widely ovate, well-developed 500–600 μm long and 400–700 μm wide (larger leaves commonly wider than longer), bifid, divided by U-shaped sinus into two prominently acute lobes, terminating by 1–3 superposed cells. Underleaves obliquely spreading, 1.1–1.5 times wider than the stem (smaller may be narrower than the stem) when looking in alive material, barely decurrent, deeply bisbifid (smaller may be simply bifid), undivided zone 1–2 cells high, rhizogenous area not developed or as 1–2 rows of smaller cells, in the slide 100–200 μm wide and 50–150 μm long. Midleaf cells greatly varying in size, (25–)30–50(–60) μm in diameter, with thin walls and vestigial concave trigones, cuticle finely papillose; cells along the margin oblong to subquadrate, 32–58 μm long and 25–38 μm wide, thin-walled, trigones very small, concave and cuticle finely verruculose to nearly smooth.

Holotype: Vietnam, Lai Châu Province, Hoang Lien Range, Hoang Lien National Park (22°20′55″ N 103°46′03″ E), 1700–1900 m a.s.l., moist cliffs in evergreen south subtropical mountain forest in deep valley, 15 March 2016, V.A. Bakalin, V-1-112-16 (VBGI).

Other specimens examined (paratypes): Vietnam, Lai Châu Province, Hoang Lien Range, Hoang Lien National Park (22°20′55″ N 103°46′03″ E), 1700–1900 m a.s.l., moist cliffs in evergreen south subtropical mountain forest in deep valley, 15 March 2016, V.A. Bakalin V-1-12-16 (VBGI), Lao Cai Province, Hoang Lien Range, Hoang Lien National Park (22°21′20″ N 103°47′33″ E), 1520 m a.s.l., evergreen south subtropical mountain forest in deep valley, partly shaded moist cliff crevice, 17 April 2017, V.A. Bakalin & K.G. Klimova V-3-14-17 (VBGI), Lai Châu Province, Pu Ta Leng Range (22°24′56″ N 103°35′47″ E), 2347 m a.s.l., partly shaded moist clayish trail cut on slope in evergreen south subtropical mountain forest in stream valley, 27 March 2018, V.A. Bakalin & K.G. Klimova V-7-58-18 (VBGI), Lao Cai Province, Hoang Lien Range, Hoang Lien National Park, southern slope of Phan Xi Pan Peak (22°18′32″ N 103°46′44″ E), 2670 m a.s.l., scattered *Abies delavayi* with many rhododendron trees and *Sinobambusa* in narrow valley, partly shaded moist clayish soil on steep slope, 4 April 2018, V.A. Bakalin & K.G. Klimova V-19-10-18 (VBGI), Lai Châu Province, Banh Hang Stream valley (22°30′00.38″ N 103°33′29.98″ E), 1509 m a.s.l., wide valley in evergreen south subtropical mountain forest, steep slope to trail, open moist clayish soil in trail cut, 8 April 2019, V.A. Bakalin & K.G. Klimova V-16-3-19 (VBGI), Sơn La Province, Xuan Nha Nature Reserve, area of Long Sap Army Gate, bottom of Dinh Pha Luong Mt. (20°41′00.9″ N 104°37′28.5″ E), 1200–1400 m a.s.l., somewhat disturbed tropical forest with the trail across steep rocky slope with sandstone enriched with calcium carbonate, partly shaded moist clayish soil in trail cut, 11 April 2019, V.A. Bakalin & K.G. Klimova V-19-9-19 (VBGI), Hà Giang Province, vicinity of Lung Tao Village, Tay Con Linh Range, Tay Con Linh Nature Reserve (22°47′30.3″ N 104°49′28.5″ E), 1503 m a.s.l., evergreen south subtropical mountain forest on SE-facing slope, open moist clayish trail cut, 20 March 2020, V.A. Bakalin & K.G. Klimova V-2-2-20 (VBGI), (22°47′50.3″ N 104°48′40.9″ E), 2191 m a.s.l., evergreen south subtropical mountain forest on SE-facing slope, partly shaded moist clayish soil near stream, 21 March 2020, V.A. Bakalin & K.G. Klimova V-11-36-20 (VBGI).

Illustrations in present paper: Figure 12 and Figure 13.

***Calypogeia pseudocuspidata*** Bakalin, Frank Müll. et Troizk., ***sp. nov.***

Description. Plants are prostrate, green when fresh and brown in the herbarium, 1.4–2.5 mm wide and 10–30 mm long, rigid, loosely attached to the substratum, forming loose patches. Rhizoids nearly absent or very rare, colorless to brownish. Stem cross section transversely ellipsoidal, 140–250 μm high and 200–350 μm wide, outer walls thin, radial walls thinner than external, inward cell walls become thicker, in outer layer 25–30 μm in diameter, inward cells larger, to 50 μm in diameter, with small, sometimes indistinct trigones. Leaves contiguous to subimbricate (overlapping to 1/3 of above situated leaf), obliquely inserted, convex, with apex turned ventrally, when flattened in the slide widely ovate to triangular ovate, well-developed 900–1300 μm long and 700–1200 μm wide (larger leaves are wider than longer), not decurrent dorsally and not or slightly ventrally, apex acute to acuminate and rarely short bicuspidate (then divided by small, U-shaped sinus). Underleaves obliquely spreading, 1.1–1.3 of stem width, shortly decurrent, divided by widely V- to U-shaped sinus into two lobes with or without additional teeth (in the vast majority of cases additional teeth very obscure or absent), undivided zone 2–4 cells high, rhizogenous area not developed or as 1–2 rows of small cells. Cells in the midleaf subisodiametric, 32–74 × 24–40 μm, thin-walled, with small but distinct concave trigones, cuticle smooth; cells along the margin are much larger than inward, 45–79 μm long and 25–41 μm wide, with thickened cell walls and noticeable thick external wall and cuticle is smooth throughout.

Holotype: Vietnam, Lao Cai Province, Sa Pa District, San Sa Ho Commune, Hoang Lien Range, Hoang Lien National Park, open moist cliff near stream in south subtropical forest in the stream valley near waterfall (22°20′22.6″ N 103°46′40.2″ E), 2014 m a.s.l., 5 April 2018, V.A. Bakalin & K.G. Klimova, V-21-17-18 (VBGI).

Other specimen examined (paratype): Myanmar, Hponyin Razi (27°36′46.6″ N 96°58′54.5″ E), 2371 m a.s.l., *Magnolia*-Fagaceae-bamboo forest, 02 November 2013, G. Miehe, P.K. Kine, L. Shein, M. Kyaw, P. Ma & S. Lan Wan, 13-081-094 g (DR, VBGI, HSNU).

Illustrations in present paper: Figure 14, Figure 15 and Figure 16A–F,H.

***Calypogeia pseudosphagnicola*** Bakalin, Troizk. et Maltseva, sp. nov.

Description. Plants are more or less soft, 1.2–1.8 mm wide and 10–20 mm long, closely attached to the substratum, whitish greenish to whitish brownish, as solitary pants over Sphagnum. Rhizoids abundant, obliquely spreading, separate, brownish. Stem cross section of well-developed plants transversely ellipsoidal, ca. 200 × 300 μm, with prominently thickened walls and moderate in size, concave trigones, (13–)25–32 μm in diameter, in the ventral side commonly smaller than in the dorsal one. Leaves contiguous to subimbricate, obliquely inserted, plane to slightly convex, with apices turned to the ventral side, not decurrent dorsally and ventrally, when flattened in the slide widely ovate, well-developed leaves 800–900 × 900–1100 μm, distinctly wider than longer, apex obtuse to somewhat acute, never divided. Underleaves narrowly spreading to nearly appressed to the stem, 1.3–2.0 times wider than the stem when looking in alive material, not decurrent, in the slide transversely ellipsoidal, divided by U- to widely V-shaped sinus into two lobes, without lateral teeth, undivided zone 3–4 cells high, rhizogenous area well-developed and 4–5 rows of small cells, 300–360 × 500–600 μm. Midleaf cells with somewhat thickened cell walls and moderate in size, concave trigones, 32–58 × 32–50 μm; cells along margin subisodiametric, with thickened walls and moderate in size to large concave trigones, 35–50 × 30–38 μm; cuticle smooth throughout.

Holotype: Russia, Russian Far East, Khabarovsk Territory, Tardoki-Yani Range, ca. 1 km westward of Tardoki-Yani Mt. peak (48°53′16.9″ N 138°02′52.8″ E), 1940 m a.s.l., moist cliff crevice in open place in steep N-facing slope with cliffs, 24 August 2013, V.A. Bakalin, Kh-40-28-13 (VBGI).

Other specimen examined (paratype): Russia, Russian Far East, Khabarovsk Territory, Baidzhalsky Range, Yarap River middle course, mountain range at the right side of the river, in the upper course of Kamenistyi Stream (50°20′44″ N 134°39′42″ E), 1640 m a.s.l., N-facing cliffs at the ridge line, open moist cliff crevice, 9 August 2016, V.A. Bakalin Kh-23-2-16 (VBGI).

Comment: The species is distinct from *C. sphagnicola* in its contiguous to subimbricate leaves, larger leaf cell size and large, distinctly wider leaves than stem underleaves. The species is also morphologically similar to *C. muelleriana*, which differs in its thickened cell walls and larger trigones in the midleaf and along the leaf margin. The species is distinct among other congeners due to prominently thickened walls in the stem cross section.

Illustrations in present paper: Figure 16G and Figure 17.

***Calypogeia neogaea*** (R.M. Schust.) Bakalin, Conserv. Biodivers. Kamchatka Coastal Waters, Proc. VII Int. Sci. Conf., Petropavlovsk-Kamchatsky, 28–29 November 2006 9, 2007.

Basionym: *Calypogeia fissa* subsp. *neogaea* R.M. Schust., Hepat. Anthocerotae N. Amer. 2: 169, 1969.

Description (based on specimen examined). Plants are pale greenish (the same in herbarium), somewhat glistening, not curly when dry, merely rigid, prostrate, in loose mats, loosely attached to the substratum, 1.6–3.0 mm wide and 15–25 mm long, in loose patches. Rhizoids abundant, erect spreading, brownish, separated. Stem cross section transversely ellipsoidal, well-developed ca 150 × 300 μm, with subequal to merely unequal thickened walls, outer cells slightly larger, 22–25 × 12–23 μm, inner cells 12–25 μm in diameter, with concave, small to vestigial trigones. Leaves obliquely to subhorizontally inserted, dorsally subtransversely inserted to arcuately inserted, concave-canaliculate, contiguous, when flattened in the slide triangular-ovate, 900–1300 × 650–1000 μm, distinctly longer than wider, apex obtuse to almost rounded, rarely bifid. Underleaves not decurrent, spreading, 1.1–1.5 times wider than the stem when looking in alive material, in the slide transversely ellipsoidal to suborbicular, 280–300 × 360–500 μm, bifid, divided by U-shaped sinus into two acute lobes, lateral additional teeth absent, unclear to distinct, although blunt, undivided zone two cells high. Midleaf cells with somewhat thickened walls, trigones small, concave, cells oblong, 45–70 × 27–38 μm; cells along margin with somewhat thickened walls, subisodiametric to oblong, with moderate in size trigones adjacent to the external side, 30–50 × 25–38 μm.

Specimen examined: U.S.A., Missouri State, Sainte Genevieve County, Hickory Canyons Natural Area, along Hickory Creek north of the loop trail (37°52′35″ N 90°18′18″ W), 300 m a.s.l., fine soil in the cave, 26 October 2014, V.A. Bakalin US-41a-77-14 (VBGI).

Illustrations in present paper: Figure 18.

***Calypogeia kamchatica*** Bakalin, Troizk. et Maltseva, ***sp. nov.***

Description. Plants are somewhat glistening, translucent, yellowish greenish to whitish yellowish, curly when dry, very soft, in loose mats, loosely attached to the substratum and 1.5–2.2 mm wide and 15–30 mm long. Rhizoids a few to nearly absent, if present, commonly appressed to the underleaves or obliquely spreading, separated. Stem cross section transversely ellipsoidal, well-developed ca 230 × 300 μm, outer cells distinctly smaller than inner, varying from 17 μm in diameter in the ventral side to 25 μm in diameter in dorsal side, with small to vestigial trigones, outer wall thin to thickened (in the ventral side), inner cells 30–40(–50) μm in diameter, with vestigial trigones and thin-walled. Leaves obliquely to very obliquely inserted, nearly plane to slightly convex and with leaf apex slightly turned to the ventral side, nearly contiguous, not decurrent dorsally and ventrally, dorsally insertion line sometimes arcuate, leaves in the slide widely ovate, 800–1100 × 600–1100 μm (small leaves are commonly longer than wider), leaf apex acute, rarely very short bifid. Underleaves distinctly decurrent for 1/2–2/3 of stem width when looking in alive material, obliquely spreading, in the slide rounded, 1.2–2.0 times wider than the stem, 260–450 × 380–700 μm, divided by U-shaped sinus into two lobes, lateral additional teeth absent or short and blunt, undivided zone 2(–4) cells high. Midleaf cells subisodiametric to oblong, 35–75 × 30–50 μm, thin-walled and trigones virtually absent; cells along leaf margin subisodiametric, 25–38 × 20–38 μm, thin-walled, trigones vestigial, but adjacent to external wall side small and concave; cuticle is smooth throughout.

Holotype: Russia, Russian Far East, Kamchatka Territory, East Kamchatka, Nalychevo Valley, middle course of Talovaya River, Kraevedcheskie Hot Springs area (53°34′30″ N 158°50′23″ E), 460 m a.s.l., flat thermal mesotrophic swamp (t~30 °C), on mineral soil, 17 August 2015, V.A. Bakalin, K-63-1-15 (VBGI).

Other specimens examined (paratypes): Russia, Russian Far East, Kamchatka Territory, East Kamchatka, Nalychevo Valley, middle course of Talovaya River, Kraevedcheskie Hot Springs area (53°34′30″ N 158°50′23″ E), 460 m a.s.l., flat thermal mesotrophic swamp (t~30 °C) in mineral soil, 17 August 2015, V.A. Bakalin, K-63-3-15 (VBGI), thermal swamp (t~15 °C) in mineral soil in hollow K-63-4-15 (VBGI), Sakhalin Province, Kuril Islands, northern part of Iturup Island, Tsirk Bay, Tsirk River middle course (45°21′45″ N 148°37′13″ E), 23 m a.s.l., moist hollow in mesotrophic swamp, 10 September 2015, V.A. Bakalin K-70-11-15, K-70-12-15 (VBGI), Kunashir Island, middle part, swampy *Picea glehnii* forest at middle course of Serebryanka River (44°02′57″ N 145°49′22″ E), 7 m a.s.l., partly shaded moist *Sphagnum* hummock, 10 September 2018, V.A. Bakalin & K.G. Klimova K-58-7-18 (VBGI), Khabarovsk Territory, Ayano-Maisky District, upper reaches of Tugorma River near the main ridge of the Dzhugdzhur Range (56°39′18.3″ N 137°15′33.9″ E), 904 m a.s.l., mosaic of *Duschekia fruticosa* and *Pinus pumila* clumps and mountain tundra vegetation on N-facing slope to stream, open wet cliff, 29 June 2019, V.A. Bakalin & K.G. Klimova Kh-38-7-19 (VBGI).

Comment. The species differs from *Calypogeia neogaea* in a number of features, including (1) twisting appearance when plants are dry (due to very soft texture), (2) very thin-walled leaf cells (versus slightly thickened), (3) underleaves larger (to 2 times wider than the stem, versus to 1.5 times the stem width), (4) distinctly and prominently decurrent underleaves (versus not decurrent), (5) stem cross section transversely ellipsoidal in both species, although much more planar in *C. neogaea* (the same as in *C. fissa*) and (6) leaves are variable in both taxa but are more regularly ovate in *C. neogaea* versus obliquely ovate-triangular in *C. kamchatica*. Nevertheless, the variation in the species is poorly understood. The plants from several specimens collected in Bering Island, Commander Arch (not cited here because the identity was not confirmed genetically) possess variation in the height of the undivided zone between underleaf lobes that may attain 4–5 cells.

Illustrations in present paper: Figure 19, Figure 20 and Figure 21.

***Calypogeia fissa*** (L.) Raddi, Jungermanniogr. Etrusca 33, 1818.

Basionym: *Mnium fissum* L., Species Plantarum 1114, 1753.

Description (based on studied specimens). Plants are in loose mats, pale greenish to greenish brownish in herbarium, somewhat glistening, prostrate, loosely attached to the substratum, 1.5–3.3 mm wide and 15–30 mm long. Rhizoids a few to abundant, obliquely to erect spreading, brownish, separated or in unclear fascicles. Stem cross section transversely ellipsoidal, in well-developed plants ca. 180 × 370 μm, cell walls thin, but in the ventral side cell walls commonly thickened in two rows, outer cells the same size with inner or slightly smaller, 20–32 μm in diameter, inner cells to 38 × 48 μm, trigones very small. Leaves obliquely inserted, not decurrent in both sides, contiguous to subimbricate, nearly plane to slightly convex, sometimes with apex turned ventrally, when flattened in the slide obliquely ovate, 600–1500 × 600–1300 μm, slightly longer than wider, apex obtuse to rarely short bifid. Underleaves obliquely spreading, commonly decurrent for 1/4 of stem width when looking in alive material, in the slide transversely ellipsoidal, 220–300 × 350–650 μm, divided by U- to V-shaped sinus in to two acute lobes, additional lateral teeth common, blunt to acute, sometimes the same length with main lobe and then underleaves bisbifid. Midleaf cells are thin-walled, with small concave trigones, oblong, 50–85 × 45–63 μm; along margin 38–80 × 30–50 μm, thin-walled, with concave small trigones, trigones adjacent to external side sometimes concave.

Specimen examined: Georgia, Adjara, Mtirala National Park, upper course of Chakvistavi River, ca. 4 km upstream of Chakvistavi Village (41°40′30.1″ N 41°52′58.1″ E) 400 m a.s.l., broadleaved Colchis forest in the stream valley, cliffs in part shade, 12 May 2013, V.A. Bakalin G-12-55-13 (VBGI).

Comment. The species differs from *C. neogaea* in its softer plant texture (versus more or less stout), leaves planar to somewhat convex (versus slightly canaliculate), leaf apex sometimes turned ventrally (versus never turned), shortly decurrent underleaves (versus not or barely decurrent), underleaves commonly bisbifid (versus never or rarely bisbifid, although commonly with blunt lateral teeth on each side). Additionally, leaf cell walls are somewhat thickened in *C. neogaea* but are thin in *C. fissa*. It is worth mentioning that the variability in this parameter should be tested along with the variability in leaf cell size (because data suggest that leaf cells are larger in *C. fissa* than in *C. neogaea*).

Illustrations in present paper: Figure 22.

***Calypogeia shevockii*** Bakalin et Troizk., sp. nov.

Description. Plants are whitish greenish, prostate to ascending, freely gemmiparous, loosely attached to the substratum, somewhat opaque, in loose mats, 1.2–2.5 mm wide and 10–20 μm long and merely soft. Rhizoids abundant, obliquely to erect spreading in unclear grayish fascicles. Stem cross section transversely ellipsoidal, well-developed stem ca 120 × 180 μm, outer cells with thickened walls (not so prominently to thin-walled in the ventral side), inner cells thin-walled to slightly unequally thickened, outer cells in the dorsal side the same size with inner cells, 22–30 μm in diameter, in the ventral side outer cells 20–25 μm in diameter, trigones small to vestigial throughout. Leaves obliquely inserted, dorsally insertion line somewhat arcuate, vernally leaves distinctly decurrent for 1/2–2/3 of stem width, plane to canaliculate or concave, when flattened in the slide obliquely ovate, 800–1000 × 700–1000 μm, mostly as long as wide, apex commonly acute, although in 20–25% leaves bifid, divided by shortly U-shaped sinus. Underleaves obliquely spreading, not or barely decurrent, 1.2–1.6 as wide as the stem when looking in alive material, in the slide 200–350 × 320–450 μm, mostly bisbifid or with prominent, although sometimes blunt lateral teeth, divided by widely U-shaped sinus, undivided zone 1–2 cells high, rhizogenous area of the same height, with 2–4 rows of small cells. Midleaf cells subisodiametric to shortly oblong, with slightly thickened walls, 30–50 × 30–50 μm, trigones vestigial, concave; cells along the margin with thin to thickened walls, subisodiametric to oblong, 35–50 × 35–45 μm, external wall slightly thickened, trigones adjacent to external wall moderate in size; cuticle smooth throughout. Gemmae abundant on attenuate shoot tips, 1–2-celled, commonly united into chains, spherical to ellipsoidal, with one papilla on the end, 20–25 μm in diameter of rounded and 20–38 × 18–25 μm if ellipsoidal.

Holotype: U.S.A., California, El Dorado County, Eldorado national forest (38°45′07.6″ N 120°26′00.0″ W), 4140 ft. alt., mixed conifer-hardwood forest with *Cornus nuttallii*, *Acer marcophyllum* and *Aralia* at hillside seeps of roadbank, 3 June 2017, J.R. Shevock 50287 (CAS, under *Calypogeia fissa*; duplicate in VBGI).

Comment. Morphologically, this species resembles small-sized, pale-colored *Calypogeia yoshinagana* (which explains the original identification of the specimen as *C. fissa*), which differs in its smaller, more deeply divided underleaves, commonly bifid leaf apex and pale coloration. Other morphologically similar species are *Asperifolia indosinica* and *A. arguta*, from which the species differs in its smooth leaf cuticle, outer cells in stem cross section similar to those inward (versus outer cells distinctly larger) and smaller leaf cells. In *trn*G topology, the species occupies a basal position to all *Calypogeia* s.l., except *Asperifolia*.

Illustrations in present paper: Figure 23.

***Calypogeia pseudointegristipula*** Bakalin, Troizk. et Maltseva, sp. nov.

Description. Plants are green to yellowish green in fresh condition and green brownish in herbarium, prostrate, closely attached to the sunstratum, 2–3 mm wide and 10–20 mm long, more or less rigid and opaque to somewhat glistening. Rhizoids abundant, obliquely spreading, in unclear fascicles. Stem cross section transversely ellipsoidal, in well-developed sems ca 350 × 600 μm, outer cells thin-walled in the ventral side, slightly thickened in the dorsal side and distinctly thick in the inner part, trigones small, concave, outer cells 20–30 μm in diameter, inner cells 25–40 × 25–30 μm. Leaves imbricate, obliquely inserted, dorsally insertion line arcuate, ventrally not or shortly decurrent, slightly convex or nearly plane, but with apex turned to the ventral side, when flattened on the slide widely obliquely ovate, 900–1400 × 100–1800 μm, distinctly wider than longer, apex rounded to somewhat obtuse. Underleaves appressed to the stem, decurrent for 1/3–1/2 of stem width, transversely ellipsoidal, shortly emarginate to distinctly divided by semicrescentic sinus into two obtuse lobes, undivided zone 10–12 cells high, rhizogenous area well developed, 500–800 × 1000–1300 μm. Midleaf cells subisodiametric, more or less thin-walled, trigones distinct, small to moderate in size, concave, 37–60 × 33–60 μm, cuticle finely verruculose; cells along the margin with thickened external wall and other walls slightly to distinctly thickened, subisodiametric to oblong, 35–45 × 25–45 μm, trigones adjacent to external wall moderate in size, concave and cuticle loosely verruculose to nearly smooth.

Holotype: Russia, Russian Far East, Sakhalin Province, Kuril Islands, Kunashir Island, Gornaya River valley middle course (44°26′22″ N 146°12′10″ E), 220 m a.s.l., moist fallen decaying wood in part shade in the valley′s forest dominating by *Picea, Abies, Sorbus commixta, Betula*, 30 August 2018, V.A. Bakalin & K.G. Klimova K-34-34-18 (VBGI).

Other specimen examined (paratype): Russia, Russian Far East, Sakhalin Province, Sakhalin Island, Korsakovsky District, Bol′shoe Vavayskoe Lake south-east vicinity (46°34′38.7″ N 143°18′32.8″ E), 6–10 m a.s.l., *Picea glehnii* forest in the plain, mesic decaying wood in part shade, 27 September 2016, V.A. Bakalin, S-43-6-16 (VBGI).

Comment. This species is probably one of the most common taxa of the genus in the southern part of the Russian Far East (Primorsky Territory, Sakhalin Island, South Kurils) and was previously misidentified as *C. integristipula*. *C. pseudointegristipula* differs from *C. integristipula* in its (1) distinctly emarginate to divided underleaves and (2) distinct trigones in the leaf cells, commonly distinct thickened cell walls along the leaf margin and commonly fine verruculose cuticle. Here, we limited the list of specimens examined to those specimens used for molecular analysis, although the distribution of the taxa is likely much wider.

Illustrations in present paper: Figure 24 and Figure 25.

## 5. Materials and Methods

### 5.1. Morphology Comparison

The material used in the present study came from our own field research, herbarium specimens and type specimens stored in various herbaria (the most valuable are G, NICH, TNS, and VBGI). The vast majority of *Calypogeia* specimens that were collected within the last 10 years were studied under ‘fresh’ conditions for oil body parameters; in many cases, oil bodies were photographed and then described. These results have led to understandings of whether molecular genetic data correlate with the differences in oil body number, size and color. The inclusion of information on oil bodies into descriptions helps us understand if the newly revealed taxa are indeed different morphologically or are cryptic. *Calypogeia,* as a genus, does not have many gametophyte morphological features that could be used in a simple identification practice; moreover, many distinguishing traits are quantitative. The main attention was given to the following characteristics: (1) oil body features, where the most readily observed are color (colorless versus brownish, blue and purple) and the presence of a central ‘eye’ (*C. japonica* only); (2) characters of the leaf apex (rounded to acute and bidentate); (3) shape of underleaves (rounded to bifid and bisbifid); (4) shoot transparency when plants are alive (plants translucent to opaque); (5) rigidity of plants (rigid to soft); (6) leaf three-dimensional shape (planar, convex, concave, with apex turned to the ventral side or not, undulate along margin, etc.); (7) the undivided zone in the underleaf between the underleaf base (above microcellous zone providing rhizoids) and the sinus (a middle sinus of three, if underleaves are bisbifid); (8) comparative size of outer and inner cells in the stem cross section (larger that inward or nearly the same); (9) comparative size and shape of cells along the leaf margin (larger than inside, swollen, elongated along margin, with thickened cell walls); (10) midleaf cell size; (11) leaf surface feature (smooth, versus papilllose or verruculose cuticle); and (12) the presence of trigones in the midleaf (actually they are always present, but sometimes are very small) and the thickness of leaf cell walls.

All studied specimens had preliminary names prior to molecular phylogenetic studies. In addition, we had a series of specimens with apparently the same morphological parameters, and some of them were selected for molecular genetic comparison. Since the taxonomic position of specimens was sometimes altered when the molecular analysis was performed, the names were corrected. For instance, all accessions of *C. kamchatica* were originally named *C. neogaea,* and when the necessity to describe the new taxon became obvious, the names were changed.

### 5.2. DNA Isolation, Amplification, and Sequencing

DNA was extracted from dried liverwort tissue using the NucleoSpin Plant II Kit (Macherey-Nagel, Düren, Nordrhein-Westfalen, Germany). Amplification of chloroplast *trn*G-intron, chloroplast *trn*L–F spacer and nuclear ITS2 was performed using an Encyclo Plus PCR Kit (Evrogen, Moscow, Russia) with the primer pairs (forward and reverse) listed in Table 3. The isolated DNA was dissolved in TE buffer and stored at −20 °C. DNA concentration was measured with a Qubit fluorometer (Invitrogen, Carlsbad, CA, USA).

The polymerase chain reaction was performed in a total volume of 20 µL, including 1 µL template DNA, 0.4 µL Encyclo polymerase, 5 µL Encyclo buffer, 0.4 µL dNTP-mixture (included in Encyclo Plus PCR Kit), 13.4 µL (for *trn*G, *trn*L–F)/12.4 µL (for ITS2) double-distilled water (Evrogen, Moscow, Russia), 1 µL dimethylsulfoxide/DMSO (for ITS2) and 0.4 µL of each primer (forward and reverse, at a concentration of 5 pmol/µL). Polymerase chain reactions were carried out using the following program: 180 s initial denaturation at 95 °C, followed by 30–40 cycles of 30 s denaturation at 94 °C, 20 (*trn*L–F)—30 s (ITS2, *trn*G) annealing at 56 °C (*trn*G), 58 °C (*trn*L–F) 60 °C or 64 °C (ITS2) and 30 s elongation at 72 °C. Final elongation was carried out in one step of 5 min at 72 °C. Amplified fragments were visualized on 1% agarose TAE gels by EtBr staining and purified using the Cleanup Mini Kit (Evrogen, Moscow, Russia). The DNA was sequenced using the BigDye Terminator v. 3.1 Cycle Sequencing Kit (Applied Biosystems, Carlsbad, CA, USA) with further analysis of the reaction products following the standard protocol on the ABI Prism 3100-Avant Genetic Analyzer (Applied Biosystems, Carlsbad, CA, USA) in the Genome Center (Engelhardt Institute of Molecular Biology, Russian Academy of Sciences, Moscow).

### 5.3. Phylogenetic Analyses

For the molecular phylogenetic study, three markers were used: nuclear 5.8S rRNA and ITS2 gene and plastid *trn*L–F region and *trn*G gene. Datasets were aligned by MAFFT using the E strategy [39] and then adjusted manually in BioEdit ver. 7.2.5 [40]. The absent parts of the sequences were coded as missing.

Phylogenies were reconstructed under three criteria: maximum parsimony (MP) with Mega X [41], maximum likelihood (ML) with IQ-TREE ver. 1.6.12 [42] and Bayesian Analyses (BA) with MrBayes ver. 3.2.7 [43].

MP analysis for all datasets included 1000 bootstrap replicates, default settings were used for all other parameters, and gaps were treated as partial deletions with a site coverage cut-off of 95%.

For the ML analysis with 1000 ultrafast bootstrap replicates [42], the best fitting evolutionary models chosen according to Bayesian Information Criterion (BIC) by IQ-TREE were K3Pu + F + G4 for *trn*G, TPM2 + F + G4 for *trn*L–F and TNe+ G4 for ITS2 datasets.

Bayesian analyses were performed by running two parallel analyses using the GTR + I + G model. For all datasets, the analysis consisted of four Markov chains. Chains were run for five million generations, and trees were sampled every 500th generation. The first 2500 trees in each run were discarded as burn-in; thereafter, 15,000 trees were sampled from both runs. Bayesian posterior probabilities were calculated from the trees sampled after burn-in. The average standard deviation of split frequencies between two runs was 0.0058 for *trn*G-intron, 0.0090 for *trn*L–F and 0.0076 for ITS2.

A haplotype network was constructed by the TCS network inference method [44] using the PopART package (http://popart.otago.ac.nz/) [45]. The PopART program automatically removes from the consideration positions having at least one N or a gap.

The infrageneric and infraspecific variability of the *trn*G intron, *trn*L–F and ITS2 were quantified as the average pairwise *p*-distances calculated in Mega X using the pairwise deletion option for counting gaps. All ambiguous positions were removed for each sequence pair.

The distribution of pairwise *p*-distances between sequences was tabulated by the Assemble Species by Automatic Partitioning (ASAP) program (https://bioinfo.mnhn.fr/abi/public/asap/asapweb.html, last accessed 15 February 2022) [46] with default settings and a *p*-distance model.

## 6. Conclusions

This account is only the first step to understanding *Calypogeia* diversity in Pacific Asia using an integrative approach; currently, many taxonomic issues remain unresolved, including the distribution of the vast majority of accepted species, which remains poorly understood. The ‘geographical races’ of each single species were not found to be genetically identical in several cases. Such supposedly circumpolar distributed taxa as *C. sphagnicola*, *C. integristipula* and *C. neesiana* are substituted in Amphi-Pacific Asia (at least in its temperate insular part) by related but nevertheless different taxa. Progress was made in advancing the knowledge of the taxonomic diversity of *Calypogeia* in Amphi-Pacific Asia due to two processes: revisiting already described but commonly neglected taxa and the description of new-to-science species.

Several particular questions remain unresolved in the course of the present study, such as the identity of the specimens named *C. cuspidata* from Vietnam with ‘true’ *C. cuspidata* described from Hawaii. Another intriguing question is the morphological differentiation of two ‘races’ of *C. granulata*, where specimens found in different clades may actually belong to different species based on their genetic distances.

Despite some gaps in this research, the tendency that has been noted several times in the literature is still obvious: the supposed ‘geographical races’ of the single ‘species’ from geographically remote areas may actually belong to distinct species that can be distinguished morphologically. One of the best examples is the morphological series of *Calypogeia fissa* s.l., where at least six species actually exist (*C. tosana*, *C. yoshinagana*, *C. fissa*, *C. shevockii*, *C. kamchatica* and *C. neogaea*).

*Calypogeia* in Vietnam was nearly completely neglected for a long time. This neglect is, for instance, illustrated by the fact that this genus was first recorded in Vietnam only five years ago, in 2017 by Shu et al. [26]. Currently, 10 taxa have been confirmed in the country, all of which were found in North Vietnam. Thus, the diversity of the genus in Vietnam will likely be updated in the course of further floristic studies. As shown here, three main obstacles prevent complete knowledge of the genus in this country: (1) *Calypogeia* are mostly hygrophilic plants and are predominantly not tropical in their distribution, which requires extensive collection in upper elevations of the mountains that may be hardly accessible; (2) oil body information is strongly required to make identifications (even if preliminary) based on morphology: this implies that the collected plants should be studied fresh as soon as possible; and (3) recent research on liverwort taxonomy has shown that semicryptic taxa and geographical vicariants are not rare cases in liverworts, and thus many difficulties arise in compiling sets for molecular genetic analyses. The listed difficulties are the same as those existing for *Calypogeia* in other areas of Amphi-Pacific Asia. Four species of *Calypogeia* are reported for Myanmar (where the taxonomic diversity in this genus may be higher than that in Vietnam), and this is the only report of the genus in that country. Thus, it is quite obvious that further study of this genus in Pacific Asia will result in the discovery of new taxa and newly revealed patterns of morphological evolution within the genus.

## Figures and Tables

**Figure 1 plants-11-00983-f001:**
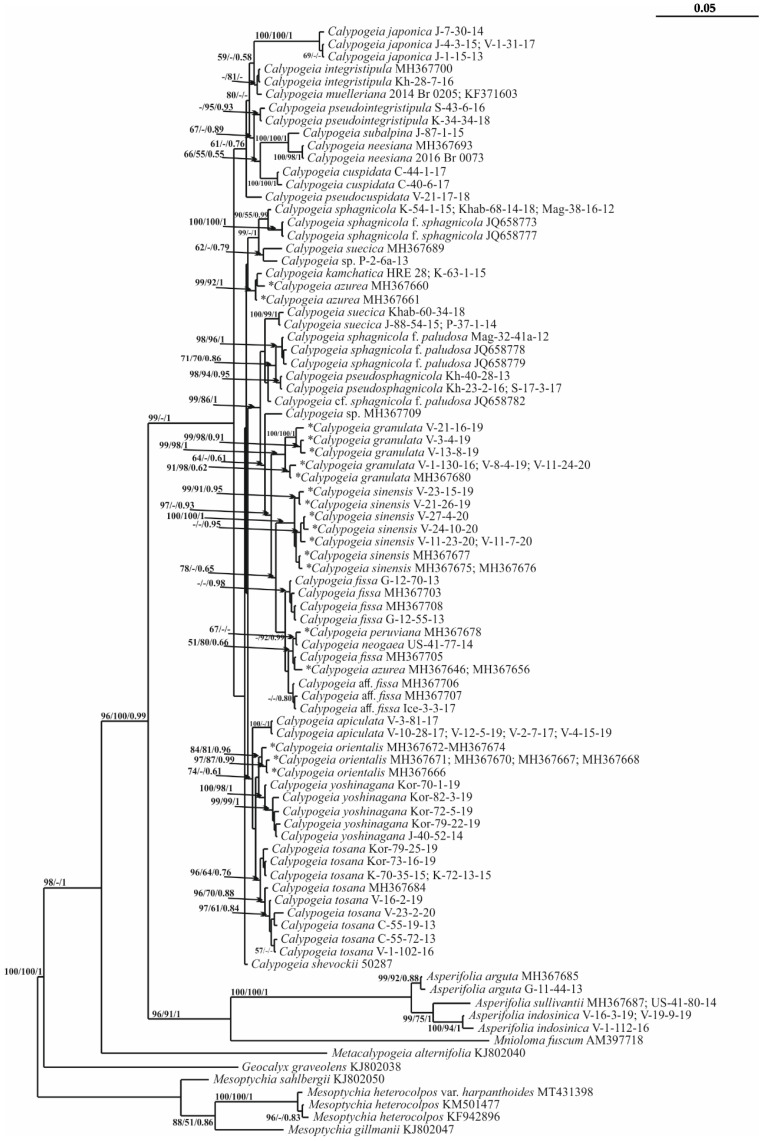
Phylogram obtained in a Bayesian analysis for the genus *Calypogeia* and related taxa based on *trn*G-intron. Bootstrap support values > 50% in the MP and ML analyses and Bayesian posterior probabilities > 50% are indicated. *Calypogeia* species possessing blue oil bodies are marked with an asterisk (*). Scale bar denotes number of nucleotide substitutions per site.

**Figure 2 plants-11-00983-f002:**
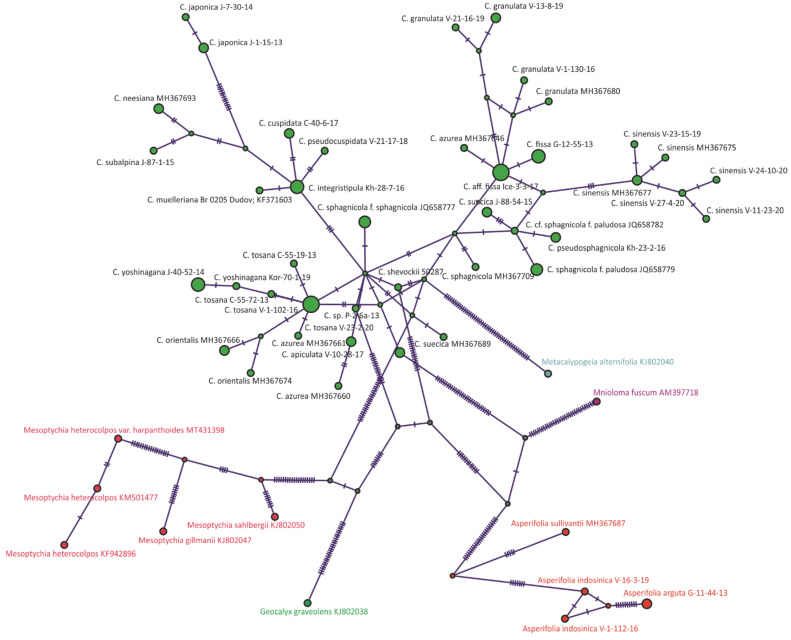
TCS haplotype network of *trn*G sequences.

**Figure 3 plants-11-00983-f003:**
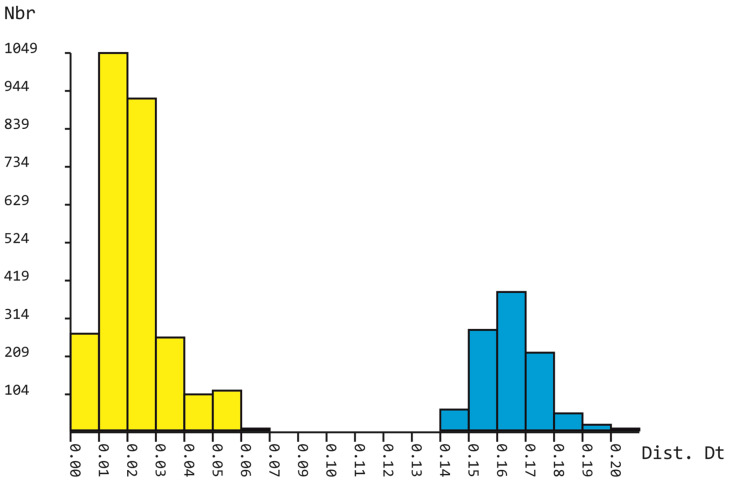
Results of Assemble Species by Automatic Partitioning (ASAP) analysis for *trn*G-intron sequences. Distribution of pairwise differences. Dt.—distance value; Nbr—number of runs.

**Figure 4 plants-11-00983-f004:**
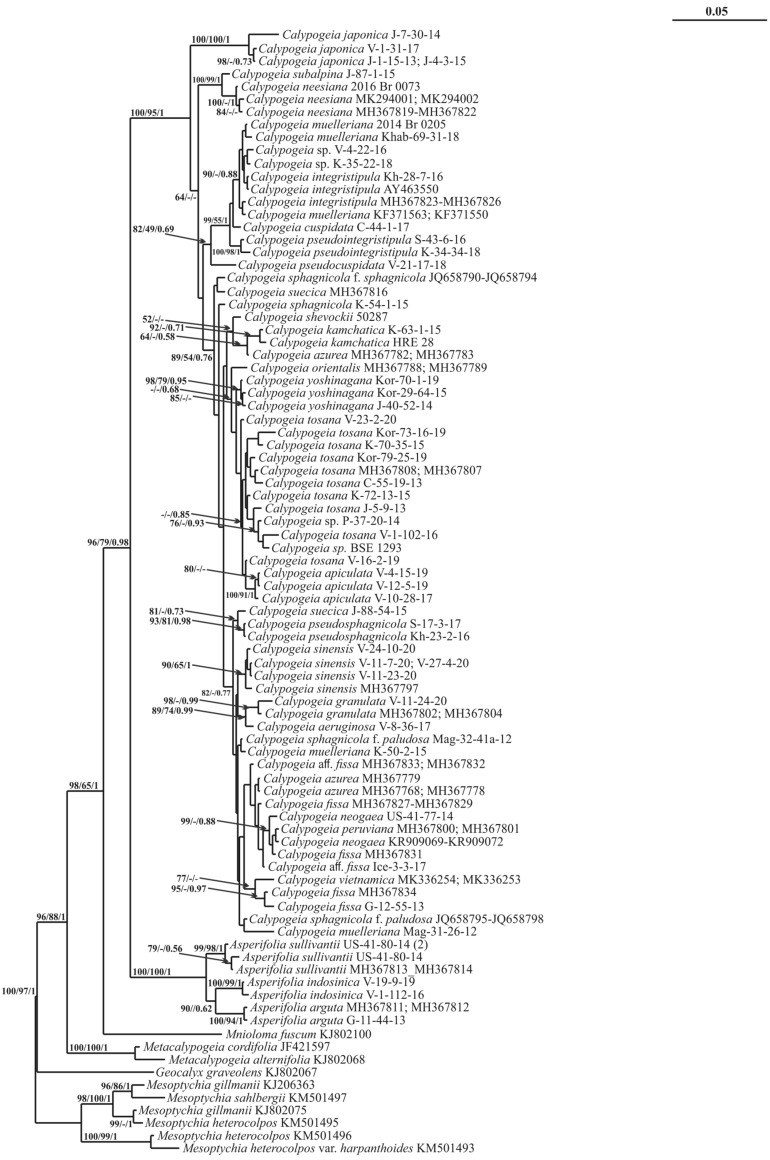
Phylogram obtained in a Bayesian analysis for the genus *Calypogeia* and related taxa based on *trn*L–F. Bootstrap support values > 50% in MP and ML analyses and Bayesian posterior probabilities > 0.50 are indicated. Scale bar denotes number of nucleotide substitutions per site.

**Figure 5 plants-11-00983-f005:**
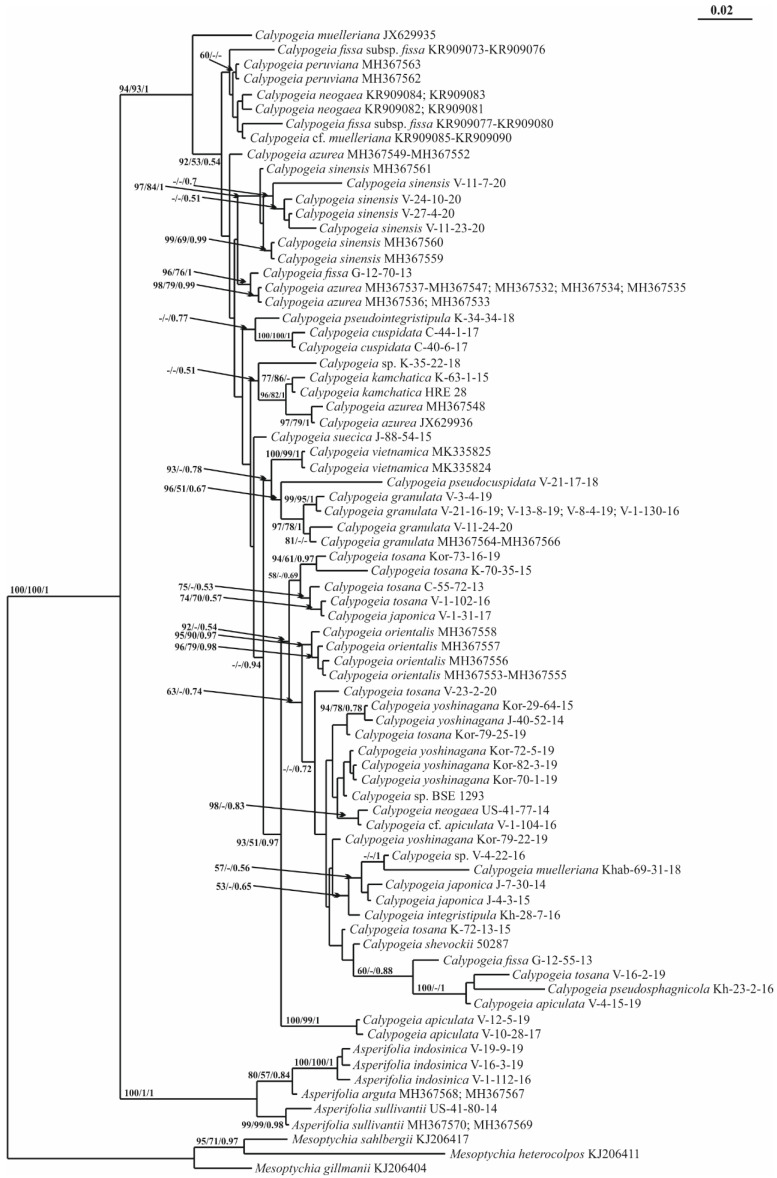
Phylogram obtained in a Bayesian analysis for the genus *Calypogeia* and related taxa based on ITS2. Bootstrap support values > 50% in MP and ML analyses and Bayesian posterior probabilities > 0.50 are indicated. Scale bar denotes number of nucleotide substitutions per site.

**Figure 6 plants-11-00983-f006:**
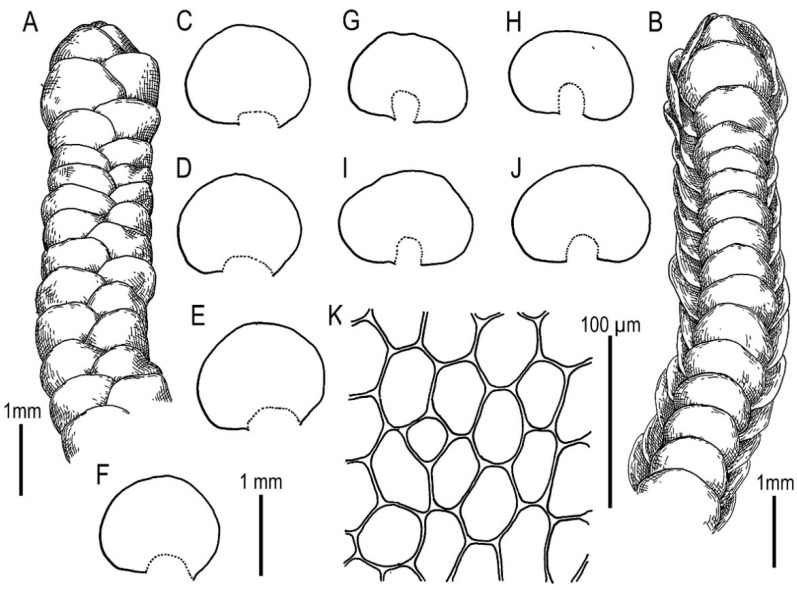
*Calypogeia aeruginosa* Mitt.: (**A**) Plant habit, fragment, dorsal view; (**B**) Plant habit, fragment, ventral view; (**C**–**F**) Leaves; (**G**–**J**) Underleaves; (**K**) Midleaf cells. All from V-8-36-17 (VBGI).

**Figure 7 plants-11-00983-f007:**
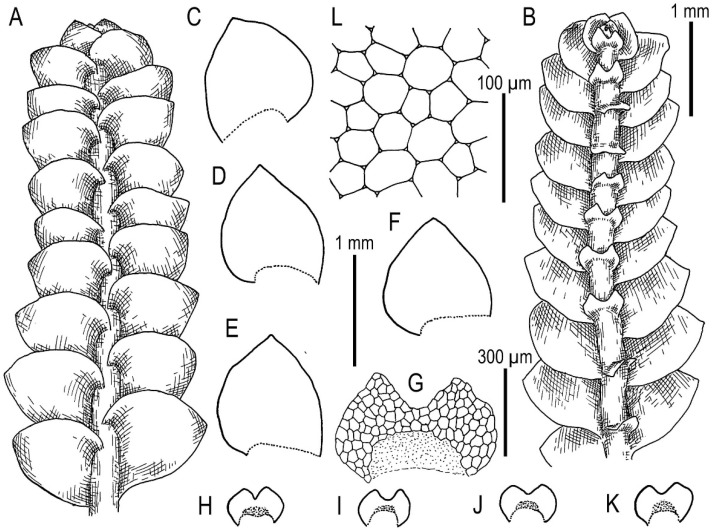
*Calypogeia apiculata* (Steph.) Steph.: (**A**) Plant habit, fragment, dorsal view; (**B**) Plant habit, fragment, ventral view; (**C**–**F**) Leaves; (**G**–**K**) Underleaves; (**L**) Midleaf cells. All from V-10-28-17 (VBGI).

**Figure 8 plants-11-00983-f008:**
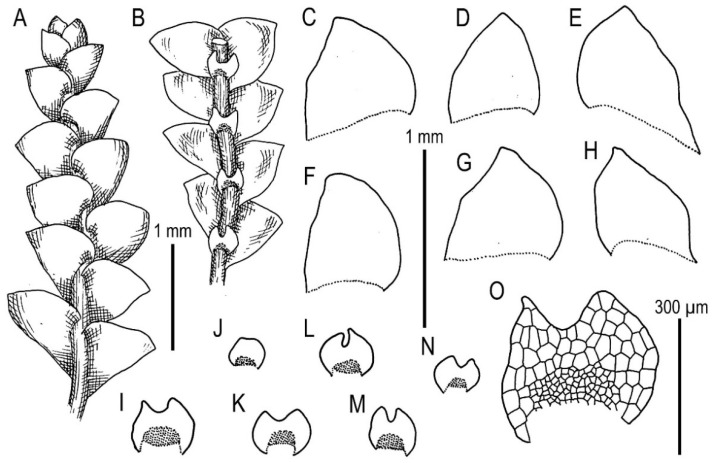
*Calypogeia cuspidata* (Steph.) Steph.: (**A**) Plant habit, fragment, dorsal view; (**B**) Plant habit, fragment, ventral view; (**C**–**H**) Leaves; (**I**–**O**) Underleaves. All from C-44-1-17 (VBGI).

**Figure 9 plants-11-00983-f009:**
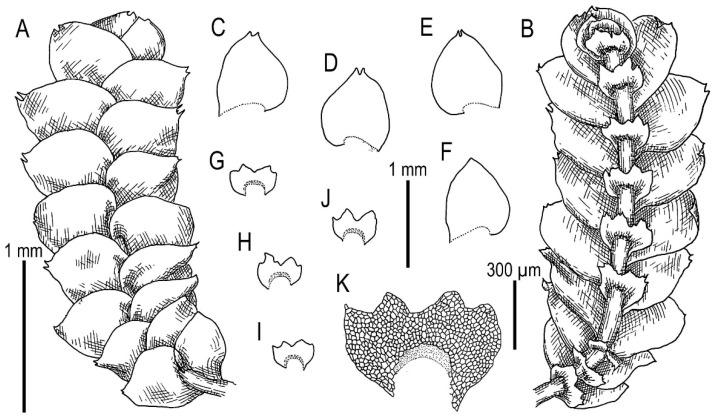
*Calypogeia granulata* Inoue: (**A**) Plant habit, fragment, dorsal view; (**B**) Plant habit, fragment, ventral view; (**C**–**F**) Leaves; (**G**–**K**) Underleaves. All from V-8-67-17 (VBGI).

**Figure 10 plants-11-00983-f010:**
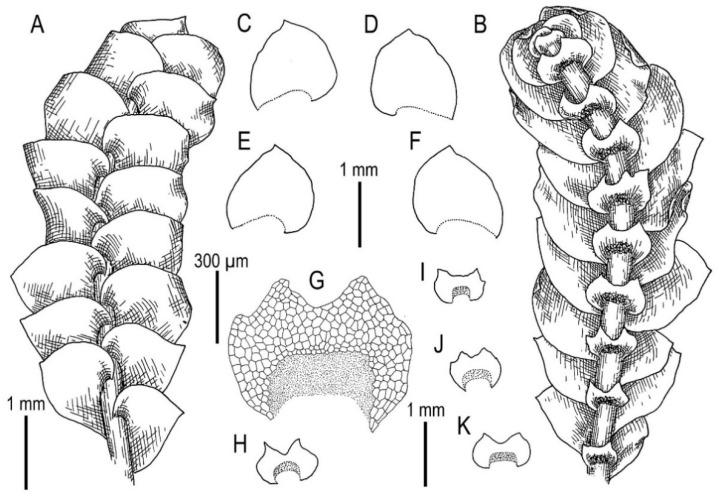
*Calypogeia sinensis* Bakalin & Buczk.: (**A**) Plant habit, fragment, dorsal view; (**B**) Plant habit, fragment, ventral view; (**C**–**F**) Leaves; (**G**–**K**) Underleaves. All from V-5-24-17 (VBGI).

**Figure 11 plants-11-00983-f011:**
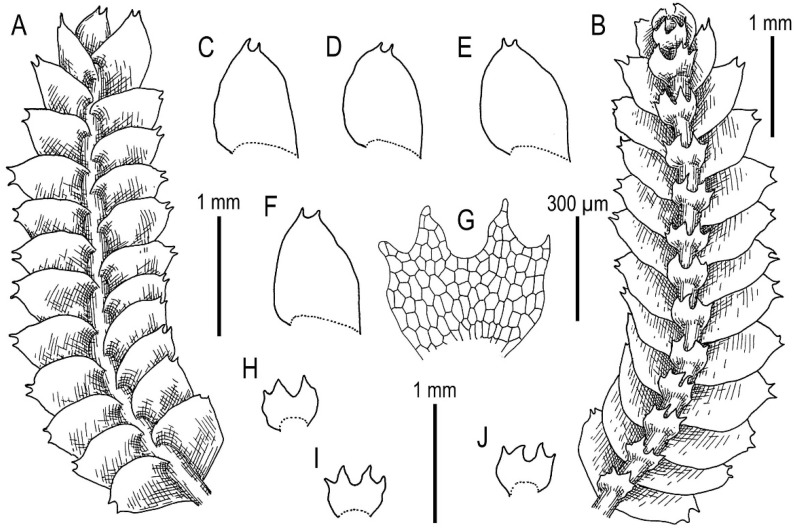
*Calypogeia tosana* (Steph.) Steph.: (**A**) Plant habit, fragment, dorsal view; (**B**) Plant habit, fragment, ventral view; (**C**–**F**) Leaves; (**G**–**J**) Underleaves. All from V-3-73-17 (VBGI).

**Figure 12 plants-11-00983-f012:**
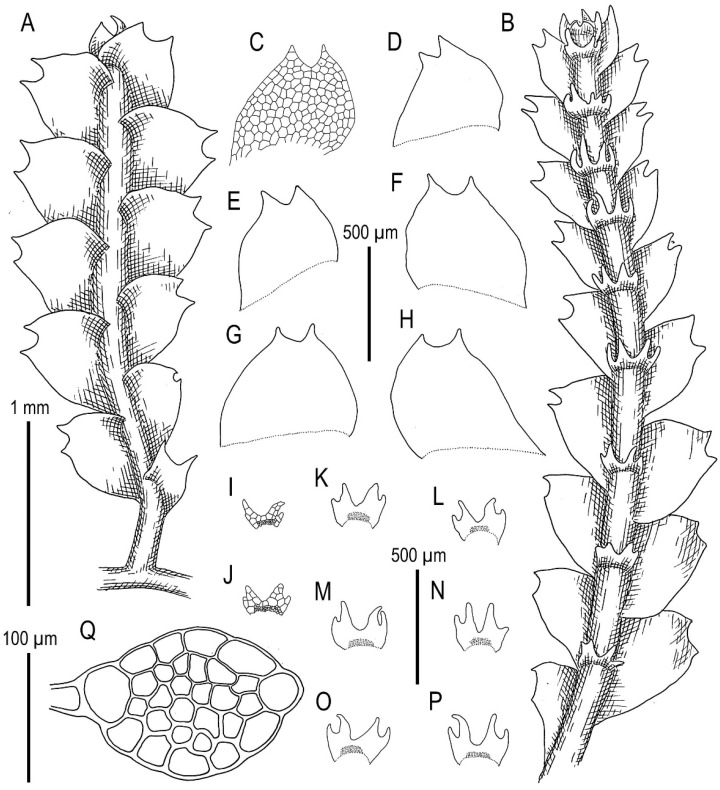
*Asperifolia indosinica* Bakalin et Troizk., *sp. nov.*: (**A**) Plant habit, fragment, dorsal view; (**B**) Plant habit, fragment, ventral view; (**C**–**H**) Leaves; (**I**–**P**) Underleaves; (**Q**) Stem cross section. All from V-1-112-16 (VBGI).

**Figure 13 plants-11-00983-f013:**
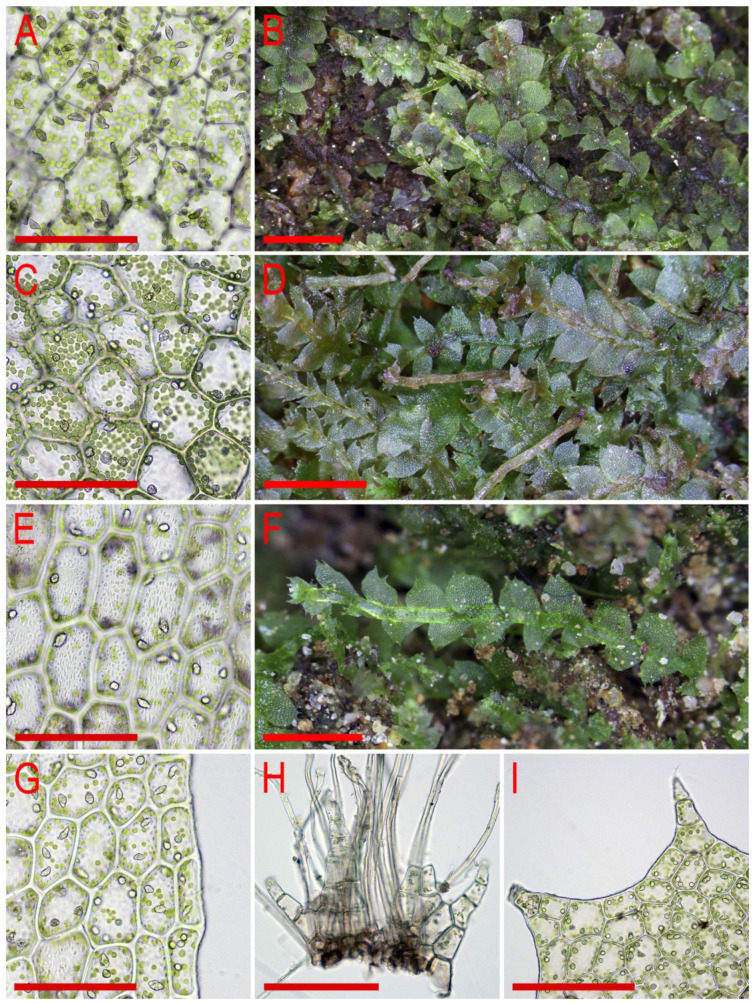
*Asperifolia indosinica* Bakalin et Troizk., *sp. nov.*: (**A**,**C**,**E**) Midleaf cells with oil bodies; (**B**,**D**) Parts of mats; (**F**) Shoot; (**G**) Leaf margin cells with oil bodies; (**H**) Underleaf; (**I**) Leaf apex cells with oil bodies. Scales: 100 µm for (**A**,**C**,**E**,**G**,**I**); 500 µm for (**H**); 1 mm for (**B**,**D**,**F**). (**A**,**B**) from V-3-14-17; (**C**,**F**) from V-2-22-16; (**D**) from V-7-58-18; (**E**,**G**–**I**) from V-19-9-19. All from VBGI.

**Figure 14 plants-11-00983-f014:**
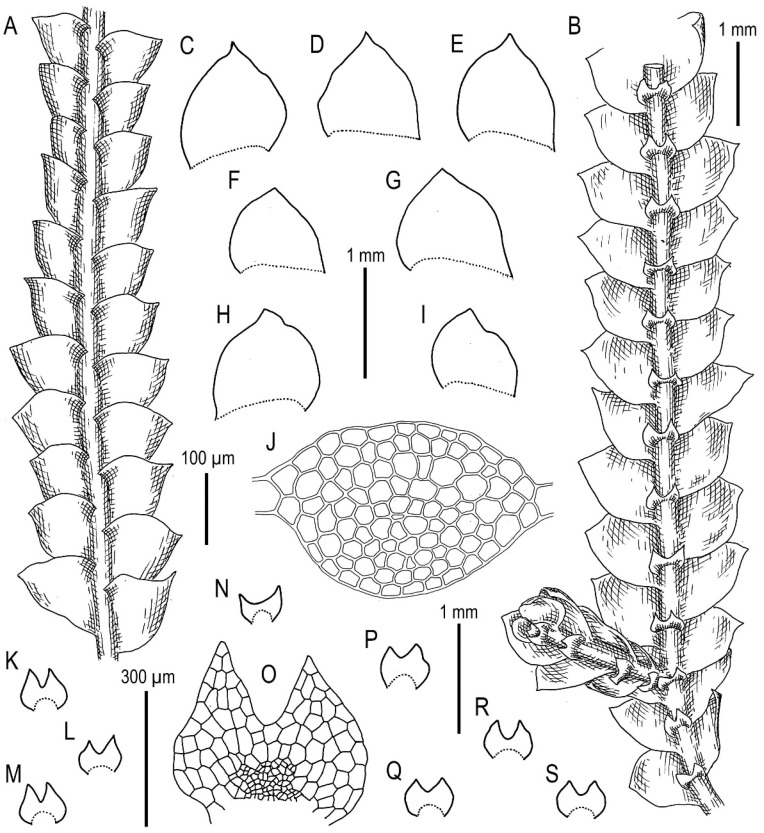
*Calypogeia pseudocuspidata* Bakalin, Frank Müll. et Troizk., *sp. nov.*: (**A**) Plant habit, fragment, dorsal view; (**B**) Plant habit, fragment, ventral view; (**C**–**I**) Leaves; (**K**–**S**) Underleaves; (**J**) Stem cross section. All from V-21-17-18 (VBGI).

**Figure 15 plants-11-00983-f015:**
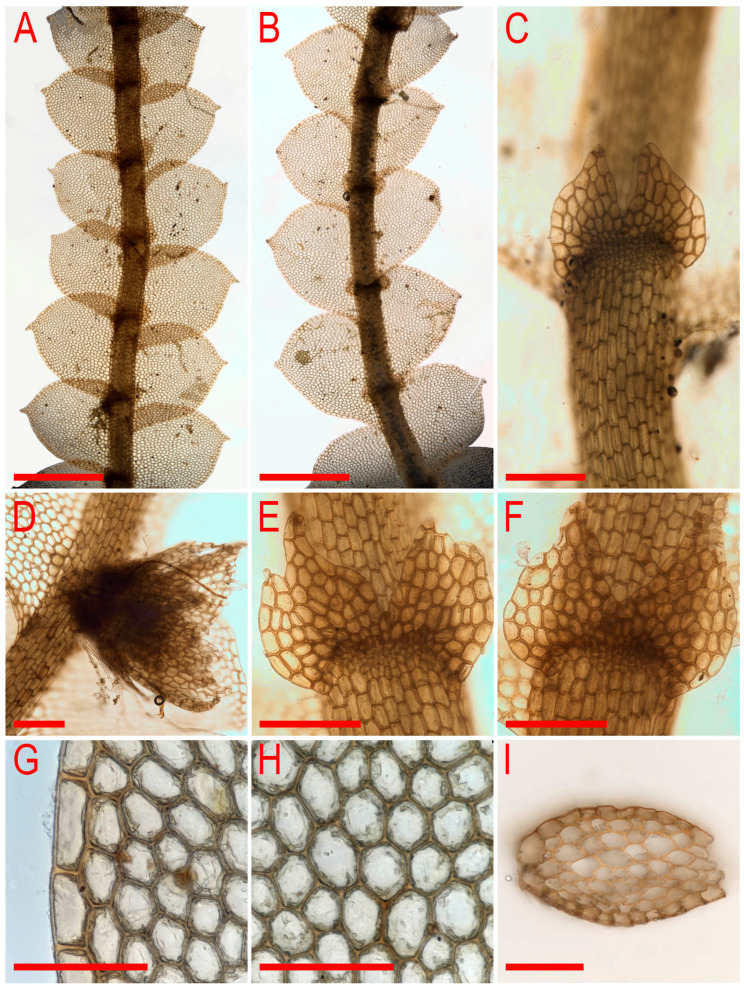
*Calypogeia pseudocuspidata* Bakalin, Frank Müll. et Troizk., *sp. nov.*: (**A**) Plant habit, fragment, dorsal view; (**B**) Plant habit, fragment, ventral view; (**C**,**E**,**F**) Underleaves; (**D**) Archegonia; (**G**) Leaf margin cells; (**H**) Midleaf cells; (**J**) Stem cross section. Scales: 1 mm for (**A**,**B**); 200 µm for (**C**–**F**); 100 µm for (**G**–**I**). All from 13-081-094-G (DR, VBGI).

**Figure 16 plants-11-00983-f016:**
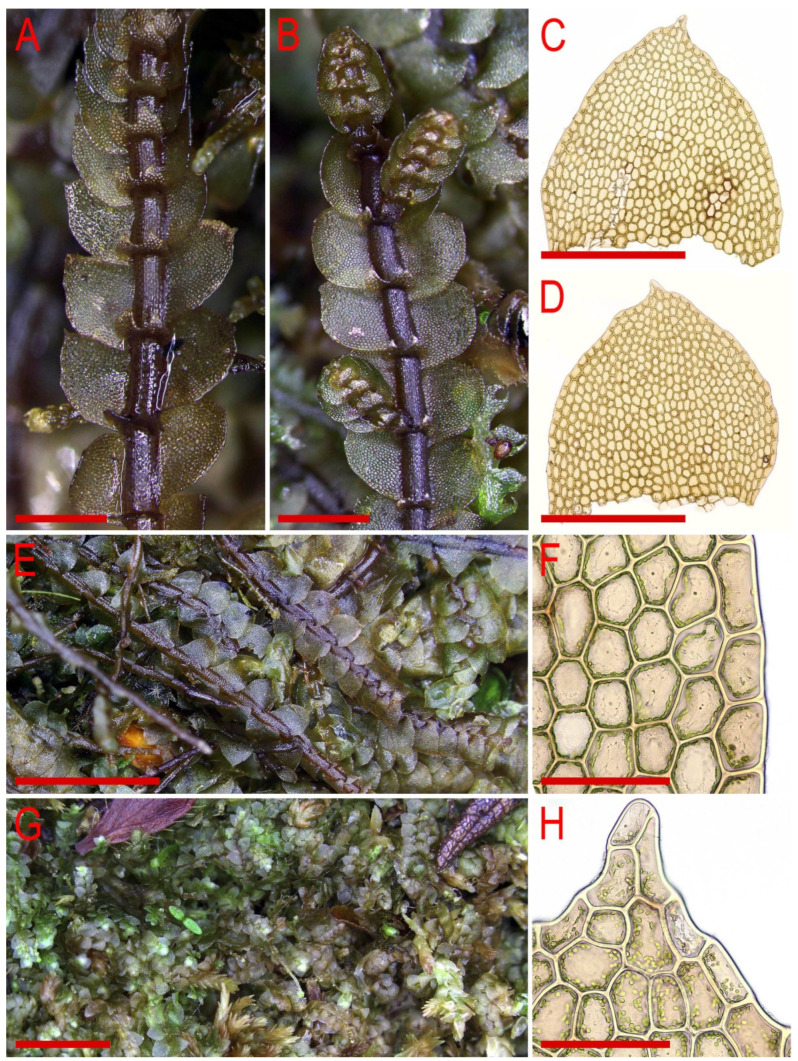
*Calypogeia pseudocuspidata* Bakalin, Frank Müll. et Troizk., *sp. nov.*: (**A**,**B**) Plant habit, fragment, ventral view; (**C**,**D**) Leaves; (**E**) Part of mat; (**F**) Leaf margin cells; (**H**) Leaf apex cells. *Calypogeia pseudosphagnicola* Bakalin, Troizk. et Maltseva, *sp. nov.*: (**G**) Mat. Scales: 1 mm for (**A**,**B**); 500 µm for (**C**,**D**); 100 µm for (**F**,**H**); 5 mm for (**E**,**G**). (**A**–**F**,**H**) from V-21-18-18; (**G**) from Kh-40-36-13. All from VBGI.

**Figure 17 plants-11-00983-f017:**
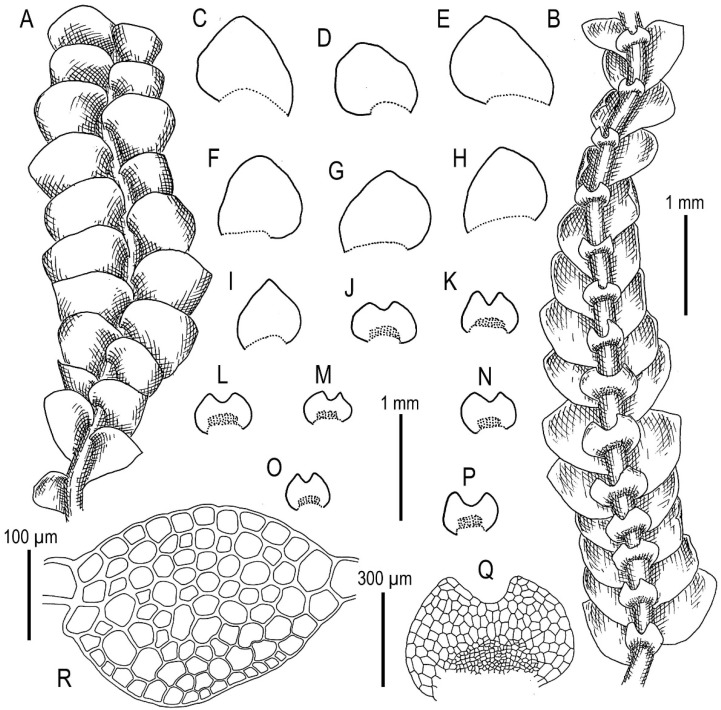
*Calypogeia pseudosphagnicola* Bakalin, Troizk. et Maltseva, *sp. nov.*: (**A**) Plant habit, fragment, dorsal view; (**B**) Plant habit, fragment, ventral view; (**C**–**I**) Leaves; (**J**–**Q**) Underleaves; (**R**) Stem cross-section. All from Kh-40-28-13 (VBGI).

**Figure 18 plants-11-00983-f018:**
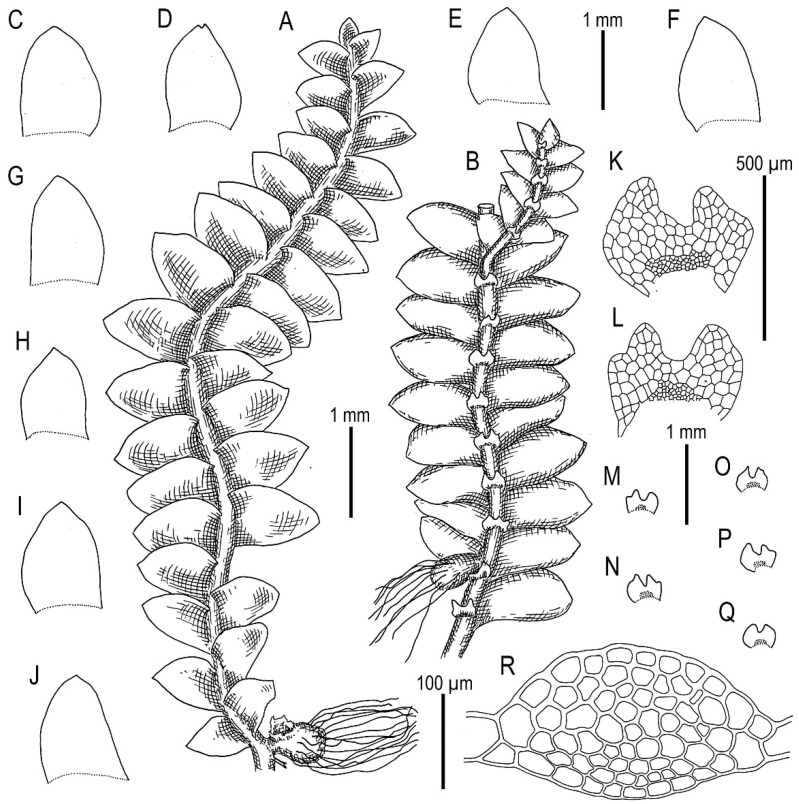
*Calypogeia neogaea* (R.M. Schust.) Bakalin: (**A**) Plant habit, fragment, dorsal view; (**B**) Plant habit, fragment, ventral view; (**C**–**J**) Leaves; (**K**–**Q**) Underleaves; (**R**) Stem cross section. All from US-41a-77-14 (VBGI).

**Figure 19 plants-11-00983-f019:**
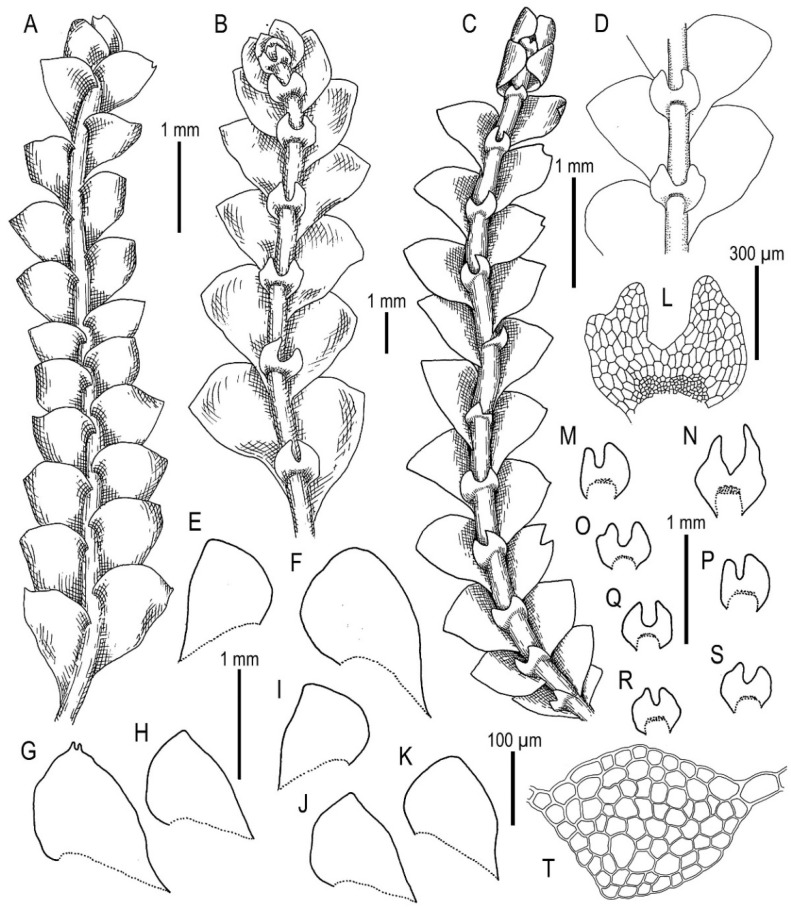
*Calypogeia kamchatica* Bakalin, Troizk. et Maltseva, *sp. nov.*: (**A**) Plant habit, fragment, dorsal view; (**B**–**D**) Plant habit, fragment, ventral view; (**E**–**K**) Leaves; (**L**–**S**) Underleaves; (**T**) Stem cross section. (**C**) From K-129-21-04 and (**A**–**T**) from K-63-1-15. All from VBGI.

**Figure 20 plants-11-00983-f020:**
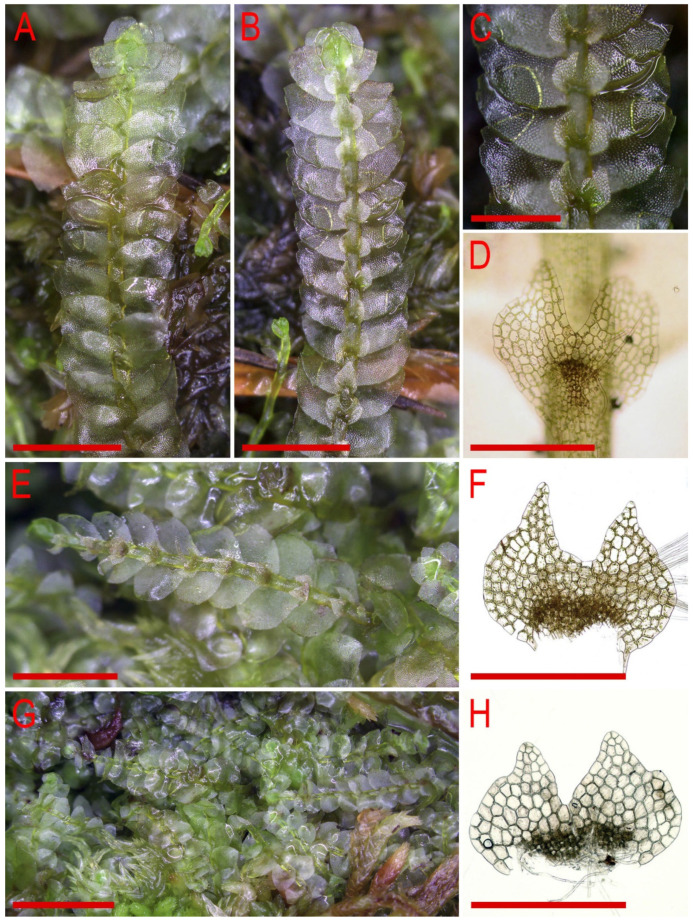
*Calypogeia kamchatica* Bakalin, Troizk. et Maltseva, *sp. nov.*: (**A**) Plant habit, fragment, dorsal view; (**B**,**C**,**E**) Plant habit, fragment, ventral view; (**D**,**F**,**H**) Underleaves; (**G**) Mat. Scales: 2 mm for (**A**,**B**,**E**); 1 mm for (**C**); 5 mm for (**G**); 500 µm for (**D**,**F**,**H**). (**A**–**C**) From K-70-11-15; (**D**) From K-63-3-15; (**E**,**G**) From K-70-12-15; (**F**,**H**) From K-58-7-18. All from VBGI.

**Figure 21 plants-11-00983-f021:**
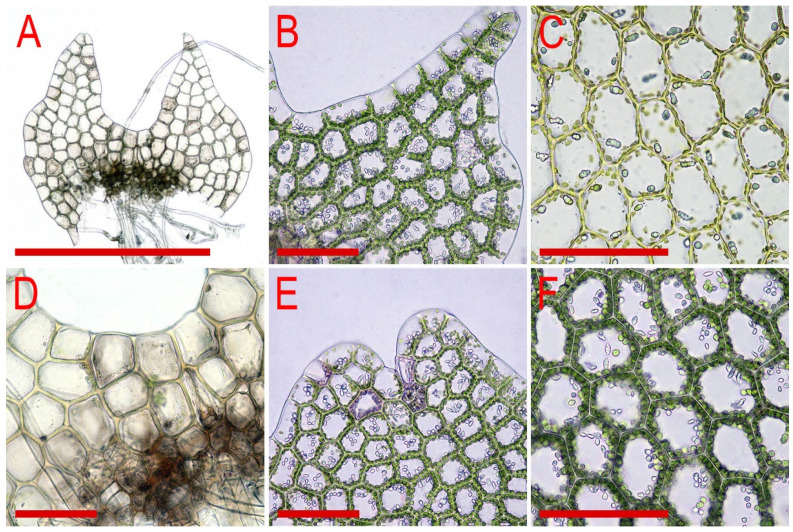
*Calypogeia kamchatica* Bakalin, Troizk. et Maltseva, *sp. nov.*: (**A**) Underleaf; (**B**) Underleaf lobe cells with oil bodies; (**C**,**F**) Midleaf cells with oil bodies; (**D**) Cells in underleaf sinus; (**E**) Leaf apex cells with oil bodies. Scales: 500 µm for (**A**); 100 µm for (**B**,**C**,**E**,**F**). (**A**,**D**) From K-58-7-18; (**B**,**E**,**F**) From Kh-38-7-19; (**C**) From K-63-4-15. All from VBGI.

**Figure 22 plants-11-00983-f022:**
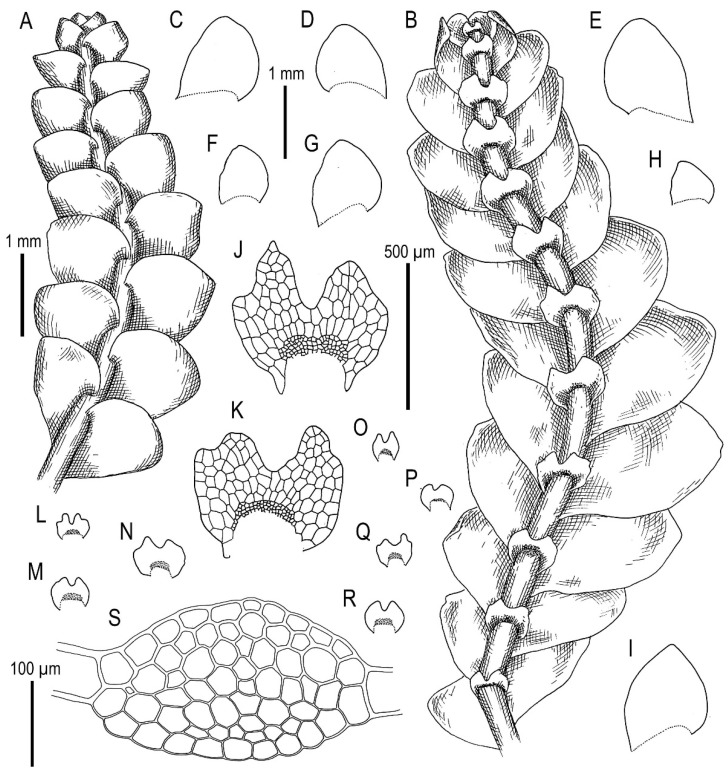
*Calypogeia fissa* (L.) Raddi: (**A**) Plant habit, fragment, dorsal view; (**B**) Plant habit, fragment, ventral view; (**C**–**I**) Leaves; (**J**–**R**) Underleaves; (**S**) Stem cross section. All from G-12-55-13 (VBGI).

**Figure 23 plants-11-00983-f023:**
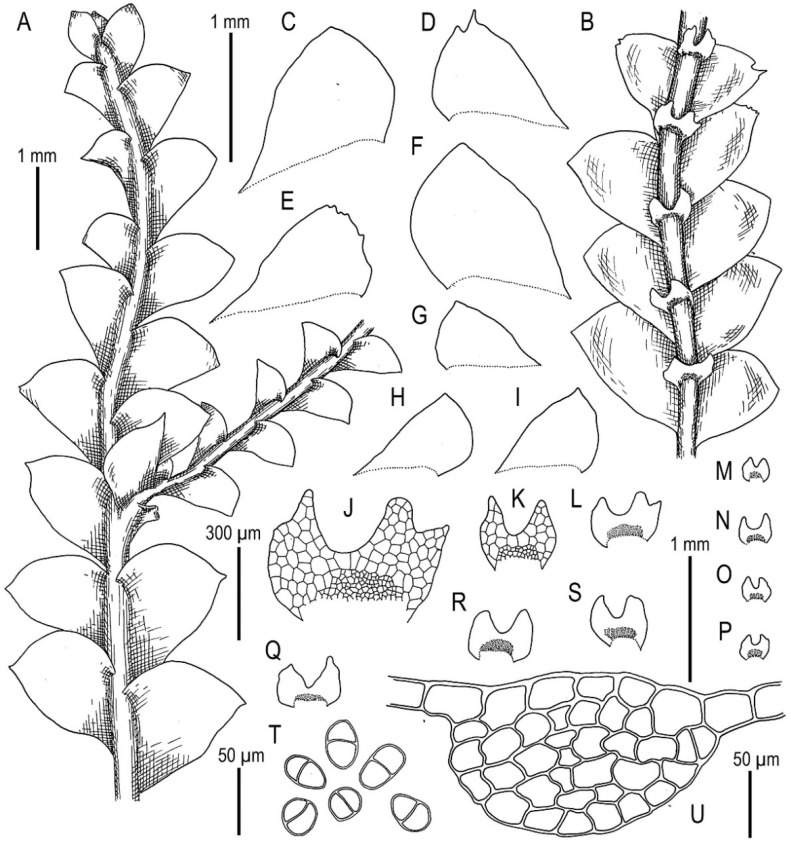
*Calypogeia shevockii* Bakalin et Troizk., *sp. nov.*: (**A**) Plant habit, fragment, dorsal view; (**B**) Plant habit, fragment, ventral view; (**C**–**I**) Leaves; (**J**–**S**) Underleaves; (**T**) Gemmae; (**U**) Stem cross section. All from Shevock 50287 (CAS, under *Calypogeia fissa*; duplicate in VBGI).

**Figure 24 plants-11-00983-f024:**
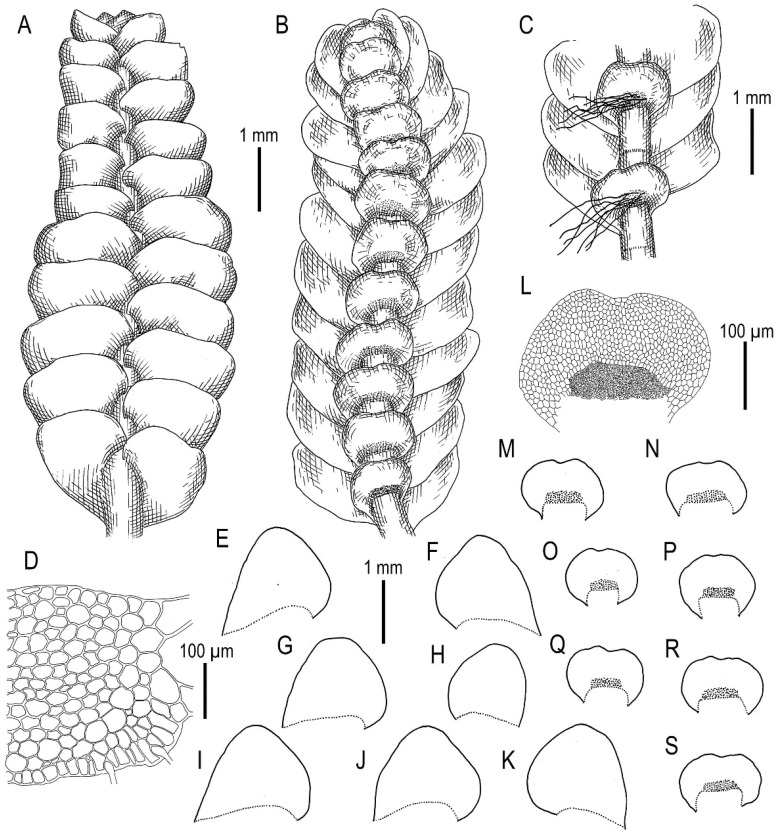
*Calypogeia pseudointegristipula* Bakalin, Troizk. et Maltseva, *sp. nov.*: (**A**) Plant habit, fragment, dorsal view; (**B**,**C**) Plant habit, fragment, ventral view; (**D**) Stem cross section, fragment; (**E**–**K**) Leaves; (**L**–**S**) Underleaves. All from K-34-34-18 (VBGI).

**Figure 25 plants-11-00983-f025:**
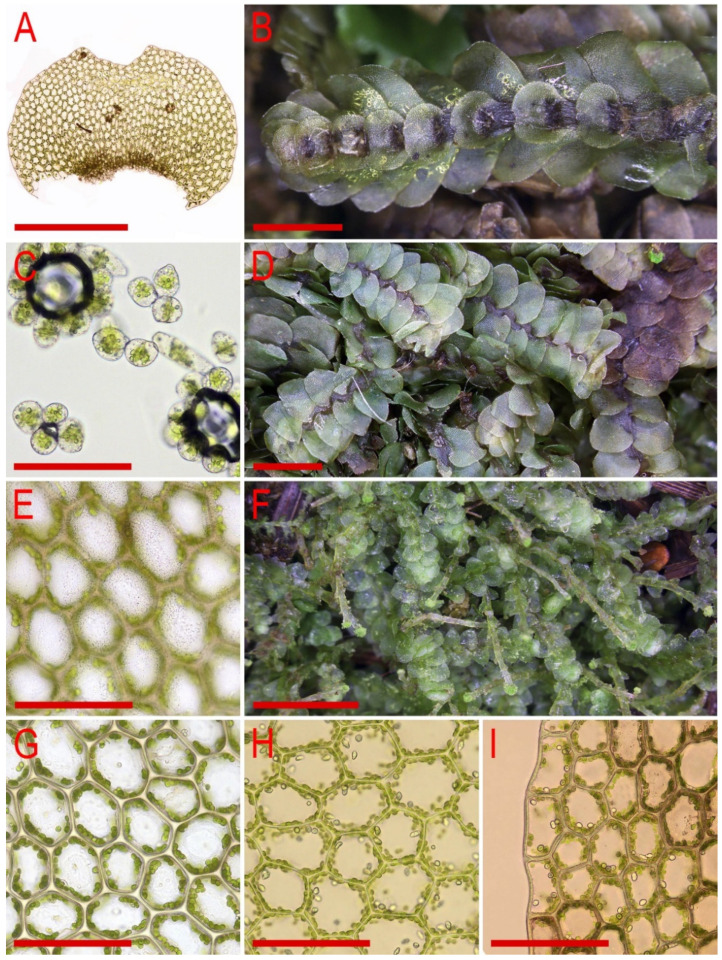
*Calypogeia pseudointegristipula* Bakalin, Troizk. et Maltseva, *sp. nov.*: (**A**) Underleaf; (**B**) Plant habit, fragment, ventral view; (**C**) Gemma with oil bodies; (**D**,**F**) Mats; (**E**) Verruculose cuticle; (**G**,**H**) Midleaf cells with oil bodies; (**I**) Leaf margin cells with oil bodies. Scales: 500 µm for (**A**); 2 mm for (**B**,**D**); 5 mm for (**F**); 100 µm for (**C**,**E**,**G**–**I**). (**A**,**B**,**D**,**E**,**G**,**I**) From K-34-34-18; (**C**,**F**,**H**) From S-43-3-16. All from VBGI.

**Table 1 plants-11-00983-t001:** The list of taxa, specimen vouchers and GenBank accession numbers.

Initial Species Name	Accepted Name	Label	GenBank Accession Number		
			ITS	*trn*L	*trn*G
*Calypogeia aeruginosa* Mitt.	*Calypogeia aeruginosa* Mitt.	Vietnam, Lao Cai Province, V.A. Bakalin & K.G. Klimova, V-8-36-17 (VBGI)	-	OL408961	-
*Calypogeia angusta* Steph.	*Calypogeia suecica* (Arnell et J.Perss.) Müll. Frib.	Japan, Honshu Island, V.A. Bakalin, J-88-54-15 (VBGI)	OL689055	OL408989	OL311475
*Calypogeia apiculata* (Steph.) Steph.	*Calypogeia apiculata* (Steph.) Steph.	Vietnam, Lao Cai Province, V.A. Bakalin & K.G. Klimova, V-2-7-17 (VBGI)	-	-	OL311465
*Calypogeia apiculata* (Steph.) Steph.	*Calypogeia apiculata* (Steph.) Steph.	Vietnam, Lao Cai Province, V.A. Bakalin & K.G. Klimova, V-3-81-17 (VBGI)	-	-	OL311466
*Calypogeia apiculata* (Steph.) Steph.	*Calypogeia apiculata* (Steph.) Steph.	Vietnam, Lao Cai Province, V.A. Bakalin & K.G. Klimova, V-10-28-17 (VBGI)	OL689064	OL409000	OL311463
*Calypogeia arguta* Nees et Mont.	*Asperifolia arguta* (Nees et Mont.) Troizk., Bakalin, Maltseva	Germany, A. Solga & D. Quandt, POZW 39075	MH367567	MH367811	MH367685
*Calypogeia arguta* Nees et Mont.	*Asperifolia arguta* (Nees et Mont.) Troizk., Bakalin, Maltseva	Spain, Galicia, A. Schäfer-Verwimp, S-V 31365	MH367568	MH367812	* MH367686
*Calypogeia arguta* Nees et Mont.	*Asperifolia indosinica* Troizk. et Bakalin	Vietnam, Lai Châu Province, V.A. Bakalin & K.G. Klimova, V-16-3-19 (VBGI)	OL689056	OL408990	OL311426
*Calypogeia arguta* Nees et Mont.	*Asperifolia indosinica* Troizk. et Bakalin	Vietnam, Sơn La Province, V.A. Bakalin & K.G. Klimova, V-19-9-19 (VBGI)	OL689065	OL409001	OL311427
*Calypogeia arguta* Nees et Mont.	*Asperifolia arguta* (Nees et Mont.) Troizk., Bakalin, Maltseva	Georgia, Adjaria, V.A. Bakalin, G-11-44-13 (VBGI)	OL689057	OL409010	OL311472
*Calypogeia arguta* Nees et Mont.	*Asperifolia indosinica* Troizk. et Bakalin	Vietnam, Lao Cai Province, V.A. Bakalin, V-1-112-16 (VBGI)	OL689079	OL409012	OL311470
*Calypogeia azurea* Stotler et Crotz	*Calypogeia azurea* Stotler et Crotz	S Poland, K. Buczkowska & A. Baczkiewicz, POZW 41776	MH367534	MH367768	MH367646
*Calypogeia azurea* Stotler et Crotz	*Calypogeia azurea* Stotler et Crotz	Germany, A. Schäfer-Verwimp, S-V 31563	MH367544	MH367778	MH367656
*Calypogeia azurea* Stotler et Crotz	*Calypogeia azurea* Stotler et Crotz	Germany, A. Schäfer-Verwimp, S-V 30124	MH367545	MH367779	* MH367657
*Calypogeia azurea* Stotler et Crotz	*Calypogeia azurea* Stotler et Crotz	North America, Canada, British Columbia, B. Aguero, POZW 42447	MH367548	MH367782	MH367660
*Calypogeia azurea* Stotler et Crotz	*Calypogeia azurea* Stotler et Crotz	Russia, Buryatiya Republic, N. Konstantinova & A. Savchenko, 20-01 (KPABG)	JX629936	* JX630063	-
*Calypogeia azurea* Stotler et Crotz	*Calypogeia azurea* Stotler et Crotz	NE Poland, K. Buczkowska & A. Bączkiewicz, POZW 41778	MH367532	* MH367766	* MH367644
*Calypogeia azurea* Stotler et Crotz	*Calypogeia azurea* Stotler et Crotz	S Poland, A. Bączkiewicz & K. Buczkowska, POZW 41390	MH367533	* MH367767	* MH367645
*Calypogeia azurea* Stotler et Crotz	*Calypogeia azurea* Stotler et Crotz	S Poland, K. Buczkowska & A. Bączkiewicz, POZW 41371	MH367535	* MH367769	* MH367647
*Calypogeia azurea* Stotler et Crotz	*Calypogeia azurea* Stotler et Crotz	S Poland, K. Buczkowska & A. Bączkiewicz, POZW 41378	MH367536	* MH367770	* MH367648
*Calypogeia azurea* Stotler et Crotz	*Calypogeia azurea* Stotler et Crotz	SE Poland, K. Buczkowska & B. Chmielewska, POZW 41925	MH367537	* MH367771	* MH367649
*Calypogeia azurea* Stotler et Crotz	*Calypogeia azurea* Stotler et Crotz	SE Poland, K. Buczkowska & B. Chmielewska, POZW 41929	MH367538	* MH367772	* MH367650
*Calypogeia azurea* Stotler et Crotz	*Calypogeia azurea* Stotler et Crotz	SE Poland, K. Buczkowska & B. Chmielewska, POZW 41948	MH367539	* MH367773	* MH367651
*Calypogeia azurea* Stotler et Crotz	*Calypogeia azurea* Stotler et Crotz	SE Poland, K. Buczkowska & B. Chmielewska, POZW 42390	MH367540	* MH367774	* MH367652
*Calypogeia azurea* Stotler et Crotz	*Calypogeia azurea* Stotler et Crotz	SE Poland, K. Buczkowska & B. Chmielewska, POZW 42373	MH367541	* MH367775	* MH367653
*Calypogeia azurea* Stotler et Crotz	*Calypogeia azurea* Stotler et Crotz	SE Poland, K. Buczkowska & B. Chmielewska, POZW 42360	MH367542	* MH367776	* MH367654
*Calypogeia azurea* Stotler et Crotz	*Calypogeia azurea* Stotler et Crotz	Austria, A. Schäfer-Verwimp, S-V 27519/A	MH367543	* MH367777	* MH367655
*Calypogeia azurea* Stotler et Crotz	*Calypogeia azurea* Stotler et Crotz	Romania, S. Ştefănut, POZW 42609	MH367546	* MH367780	* MH367658
*Calypogeia azurea* Stotler et Crotz	*Calypogeia azurea* Stotler et Crotz	Russia, Northern Caucasus, Karachai-Cherkessk Republic, N.A. Konstantinova & A.N. Savchenko, HRE 57	MH367547	* MH367781	* MH367659
*Calypogeia azurea* Stotler et Crotz	*Calypogeia azurea* Stotler et Crotz	North America, Vancouver Island, B. Aguero, POZW 42444	MH367549	MH367783	MH367661
*Calypogeia azurea* Stotler et Crotz	*Calypogeia azurea* Stotler et Crotz	USA, California Modoc Plateau, W.T. Doyle, MO 6005455	MH367550	* MH367784	* MH367662
*Calypogeia azurea* Stotler et Crotz	*Calypogeia azurea* Stotler et Crotz	USA, Washington, W. Hong, MO 5241997	MH367551	* MH367785	* MH367663
*Calypogeia azurea* Stotler et Crotz	*Calypogeia azurea* Stotler et Crotz	USA, California, W.T. Doyle, 00234919 (NYBG)	MH367552	* MH367786	* MH367664
*Calypogeia* cf. *apiculata* (Steph.) Steph.	*Calypogeia* cf. *apiculata* (Steph.) Steph.	Vietnam, Lao Cai Province, V.A. Bakalin, V-1-104-16 (VBGI)	OL689072	-	-
*Calypogeia* cf. *fissa* (L.) Raddi	*Calypogeia fissa* (L.) Raddi	Georgia, Adjaria, V.A. Bakalin, G-12-70-13 (VBGI)	OL689083	-	OL311474
*Calypogeia* cf. *integristipula* Steph.	*Calypogeia pseudointegristipula* Bakalin, Troizk. et Maltseva	Russia, Sakhalin Province, Southern Kuril Islands, Kunashir Island, V.A. Bakalin & K.G. Klimova, K-34-34-18 (VBGI)	OL689071	OL409006	OL311448
*Calypogeia* cf. *japonica* Steph.	*Calypogeia* sp.	Russia, Russian Far East, Southern Kurils, Kunashir Island, V.A. Bakalin & K.G. Klimova, K-35-22-18 (VBGI)	OL689074	OL409008	-
*Calypogeia* cf. *muelleriana* (Schiffn.) Müll.Frib.	*Calypogeia* cf. *muelleriana* (Schiffn.) Müll.Frib.	Poland, K. Buczkowska & A. Bączkiewicz, POZW 42211	KR909089	* KF371566	* KF371618
*Calypogeia* cf. *muelleriana* (Schiffn.) Müll.Frib.	*Calypogeia* cf. *muelleriana* (Schiffn.) Müll.Frib.	Poland, K. Buczkowska & A. Bączkiewicz, POZW 42220	KR909087	* KF371568	* KF371620
*Calypogeia* cf. *muelleriana* (Schiffn.) Müll.Frib.	*Calypogeia* cf. *muelleriana* (Schiffn.) Müll.Frib.	Poland, K. Buczkowska & A. Bączkiewicz, POZW 41707	KR909086	* KF371570	* KF371622
*Calypogeia* cf. *muelleriana* (Schiffn.) Müll.Frib.	*Calypogeia* cf. *muelleriana* (Schiffn.) Müll.Frib.	Poland, K. Buczkowska & A. Bączkiewicz, POZW 41706	KR909085	* KF371571	* KF371623
*Calypogeia* cf. *muelleriana* (Schiffn.) Müll.Frib.	*Calypogeia* cf. *muelleriana* (Schiffn.) Müll.Frib.	Poland, K. Buczkowska & A. Bączkiewicz, POZW 42285a	KR909088	* KF371573	* KF371625
*Calypogeia* cf. *muelleriana* (Schiffn.) Müll.Frib.	*Calypogeia* cf. *muelleriana* (Schiffn.) Müll.Frib.	USA, B.K. Andreas, B.K.Andreas dhl825	KR909090	* KF371575	* KF371627
*Calypogeia* cf. *sphagnicola* f. *paludosa* (Warnst.) R.M. Schust.	*Calypogeia* cf. *sphagnicola* f. *paludosa* (Warnst.) R.M. Schust.	S Poland, POZW 41166	-	* JQ658799	JQ658782
*Calypogeia* cf. *tosana* (Steph.) Steph.	*Calypogeia japonica* Steph.	Vietnam, Lao Cai Province, V.A. Bakalin & K.G. Klimova, V-1-31-17 (VBGI)	OL689036	OL409013	OL311467
*Calypogeia cuspidata* (Steph.) Steph.	*Calypogeia cuspidata* (Steph.) Steph.	China, Sichuan Province, V.A. Bakalin & K.G. Klimova, C-44-1-17 (VBGI)	OL689041	OL408967	OL311478
*Calypogeia cuspidata* (Steph.) Steph.	*Calypogeia cuspidata* (Steph.) Steph.	China, Sichuan Province, V.A. Bakalin & K.G. Klimova, C-40-6-17 (VBGI)	OL689038	OL409015	OL311480
*Calypogeia fissa* (L.) Raddi	*Calypogeia fissa* (L.) Raddi	Germany, A. Solga, D. Quandt, POZW 39074	-	MH367827	* MH367701
*Calypogeia fissa* (L.) Raddi	*Calypogeia fissa* (L.) Raddi	W Poland, Lubuskie Province, S. Rosadzinski, POZW 42437	-	MH367828	* MH367702
*Calypogeia fissa* (L.) Raddi	*Calypogeia fissa* (L.) Raddi	Germany, A. Schäfer-Verwimp, S-V 25448	-	MH367829	MH367703
*Calypogeia fissa* (L.) Raddi	*Calypogeia fissa* (L.) Raddi	North America, Canada, Nova Scotia, J. Macoun, POZW 3337	-	MH367831	MH367705
*Calypogeia fissa* (L.) Raddi	*Calypogeia* aff. *fissa* (L.) Raddi	Ireland, J.S. Thomson, POZW 3344	-	MH367832	MH367706
*Calypogeia fissa* (L.) Raddi	*Calypogeia* aff. *fissa* (L.) Raddi	Spain, C. Casas, POZW 3345	-	MH367833	MH367707
*Calypogeia fissa* (L.) Raddi	*Calypogeia fissa* (L.) Raddi	Hungary, L. Vajda, POZW 3347	-	MH367834	MH367708
*Calypogeia fissa* (L.) Raddi	*Calypogeia* aff. *fissa* (L.) Raddi	Iceland, Austurnssysla, V.V. Buryy, Ice-3-3-17	-	OL408960	OL311418
*Calypogeia fissa* (L.) Raddi	*Calypogeia shevockii* Bakalin et Troizk.	USA, California State, J.R. Shevock, 50287	OL689039	OL408965	OL311477
*Calypogeia fissa* (L.) Raddi	*Calypogeia fissa* (L.) Raddi	Georgia, Adjaria, V.A. Bakalin, G-12-55-13 (VBGI)	OL689037	OL408968	OL311473
*Calypogeia fissa* (L.) Raddi	*Asperifolia sullivantii* (Austin) Troizk., Bakalin, Maltseva	USA, Sainte Genevieve County, V.A. Bakalin, US-41a-80-14 (VBGI)	OL689045	OL408971	OL409018
*Calypogeia fissa* (L.) Raddi subsp. *fissa*	*Calypogeia fissa* (L.) Raddi subsp. *fissa*	Poland, S. Rosadziński, POZW 42628	KR909073	* KR909061	-
*Calypogeia fissa* (L.) Raddi subsp. *fissa*	*Calypogeia fissa* (L.) Raddi subsp. *fissa*	Poland, K. Buczkowska & J. Szweykowski, POZW 39203	KR909074	* KR909062	-
*Calypogeia fissa* (L.) Raddi subsp. *fissa*	*Calypogeia fissa* (L.) Raddi subsp. *fissa*	Poland, K. Buczkowska, POZW 42225	KR909075	* KR909063	-
*Calypogeia fissa* (L.) Raddi subsp. *fissa*	*Calypogeia fissa* (L.) Raddi subsp. *fissa*	Poland, K. Buczkowska, POZW 42227	KR909076	* KR909064	-
*Calypogeia fissa* (L.) Raddi subsp. *fissa*	*Calypogeia fissa* (L.) Raddi subsp. *fissa*	Poland, K. Buczkowska & A. Bączkiewicz, POZW 42345	KR909077	* KR909065	-
*Calypogeia fissa* (L.) Raddi subsp. *fissa*	*Calypogeia fissa* (L.) Raddi subsp. *fissa*	Poland, K. Buczkowska & A. Bączkiewicz, POZW 42205	KR909078	* KR909066	-
*Calypogeia fissa* (L.) Raddi subsp. *fissa*	*Calypogeia fissa* (L.) Raddi subsp. *fissa*	Poland, K. Buczkowska & A. Bączkiewicz, POZW 42200	KR909079	* KR909067	-
*Calypogeia fissa* (L.) Raddi subsp. *fissa*	*Calypogeia fissa* (L.) Raddi subsp. *fissa*	Poland, S. Rosadziński & K. Buczkowska, POZW 42298	KR909080	* KR909068	-
*Calypogeia fissa* subsp. *neogaea* R.M. Schust.	*Calypogeia neogaea* (R.M. Schust.) Bakalin	USA, North Carolina, B. Shaw, POZW 42620	KR909081	KR909069	-
*Calypogeia fissa* subsp. *neogaea* R.M. Schust.	*Calypogeia neogaea* (R.M. Schust.) Ba-kalin	USA, North Carolina, B. Shaw, POZW 42622	KR909082	KR909070	-
*Calypogeia fissa* subsp. *neogaea* R.M. Schust.	*Calypogeia neogaea* (R.M. Schust.) Bakalin	USA, North Carolina, B. Shaw, POZW 42625	KR909083	KR909071	-
*Calypogeia fissa* subsp. *neogaea* R.M. Schust.	*Calypogeia neogaea* (R.M. Schust.) Bakalin	USA, North Carolina, B. Shaw, POZW 42626	KR909084	KR909072	-
*Calypogeia granulata* Inoue	*Calypogeia granulata* Inoue	Vietnam, Lai Châu Province, V.A. Bakalin & K.G. Klimova, V-13-8-19 (VBGI)	OL689076	-	OL311419
*Calypogeia granulata* Inoue	*Calypogeia granulata* Inoue	Vietnam, Sơn La Province, V.A. Bakalin & K.G. Klimova, V-21-16-19 (VBGI)	OL689080	-	OL311420
*Calypogeia granulata* Inoue	*Calypogeia granulata* Inoue	Vietnam, Lai Châu Province, V.A. Bakalin & K.G. Klimova, V-3-4-19 (VBGI)	OL689084	-	OL311421
*Calypogeia granulata* Inoue	*Calypogeia granulata* Inoue	Vietnam, Lai Châu Province, V.A. Bakalin & K.G. Klimova, V-8-4-19 (VBGI)	OL689034	-	OL311422
*Calypogeia granulata* Inoue	*Calypogeia granulata* Inoue	Japan, H. Inoue, JE 18004	MH367564	MH367802	MH367680
*Calypogeia granulata* Inoue	*Calypogeia granulata* Inoue	China, Guizhou Province, V.A. Bakalin, China-56-19-13 (VBGI)	MH367566	MH367804	* MH367682
*Calypogeia granulata* Inoue	*Calypogeia granulata* Inoue	Vietnam, Hà Giang Province, V.A. Bakalin & K.G. Klimova, V-11-24-20 (VBGI)	OL689050	OL770363	OL770364
*Calypogeia granulata* Inoue	*Calypogeia granulata* Inoue	Japan, Kuroyama, H. Inoue, 2792205 (NYGB)	MH367565	* MH367803	* MH367681
*Calypogeia integristipula* Steph.	*Calypogeia integristipula* Steph.	Finland, He-Nygrèn & Piippo 1472	-	AY463550	-
*Calypogeia integristipula* Steph.	*Calypogeia integristipula* Steph.	NE Poland, K. Buczkowska & A. Bączkiewicz, POZW 41730	-	MH367823	* MH367697
*Calypogeia integristipula* Steph.	*Calypogeia integristipula* Steph.	SE Poland, K. Buczkowska & B. Chmielewski, POZW 41928	-	MH367824	* MH367698
*Calypogeia integristipula* Steph.	*Calypogeia integristipula* Steph.	NE Poland, K. Buczkowska & A. Bączkiewicz, POZW 41785	-	MH367825	* MH367699
*Calypogeia integristipula* Steph.	*Calypogeia integristipula* Steph.	NW Poland, K. Buczkowska & A. Bączkiewicz, POZW 41188	-	MH367826	MH367700
*Calypogeia integristipula* Steph.	*Calypogeia integristipula* Steph.	Russia, Russian Far East, Khabarovsk Territory, V.A. Bakalin, Kh-28-7-16 (VBGI)	OL689044	OL408970	OL311423
*Calypogeia japonica* Steph.	*Calypogeia japonica* Steph.	Japan, Honshu Island, V.A. Bakalin, J-1-15-13 (VBGI)	-	OL408974	OL311424
*Calypogeia japonica* Steph.	*Calypogeia japonica* Steph.	Japan, Shikoku Island, V.A. Bakalin, J-4-3-15 (VBGI)	OL689047	OL408975	OL311476
*Calypogeia japonica* Steph.	*Calypogeia japonica* Steph.	Japan, Kyushu Island, V.A. Bakalin, J-7-30-14 (VBGI)	OL689048	OL408976	OL311425
*Calypogeia japonica* Steph.	*Calypogeia* sp.	Vietnam, Lao Cai Province, V.A. Bakalin, V-4-22-16 (VBGI)	OL689053	OL408981	-
*Calypogeia muelleriana* (Schiffn.) Müll.Frib.	*Calypogeia muelleriana* (Schiffn.) Müll.Frib.	Russia, Perm Province, N. Konstantinova, K 367-1-04 (KPABG)	JX629935	* JX630062	-
*Calypogeia muelleriana* (Schiffn.) Müll.Frib.	*Calypogeia muelleriana* (Schiffn.) Müll.Frib.	Poland, K. Buczkowska & A. Bączkiewicz, POZW 41346	-	KF371550	* KF371602
*Calypogeia muelleriana* (Schiffn.) Müll.Frib.	*Calypogeia muelleriana* (Schiffn.) Müll.Frib.	Poland, K. Buczkowska, A. Baczkiewicz, POZW 41182	-	* KF371551	KF371603
*Calypogeia muelleriana* (Schiffn.) Müll.Frib.	*Calypogeia muelleriana* (Schiffn.) Müll.Frib.	Netherlands, J. Szweykowski, R. Gradstein, H. Greven, POZW 34160	-	KF371563	* KF371615
*Calypogeia muelleriana* (Schiffn.) Müll.Frib.	*Calypogeia muelleriana* (Schiffn.) Müll.Frib.	Russia, Russian Far East, Kamchatka Territory, V.A. Bakalin, K-50-2-15 (VBGI)	-	OL408982	-
*Calypogeia muelleriana* (Schiffn.) Müll.Frib.	*Calypogeia pseudosphagnicola* Bakalin, Troizk. et Maltseva	Russia, Russian Far East, Khabarovsk Territory, V.A. Bakalin, Kh-23-2-16 (VBGI)	OL689054	OL408983	OL311453
*Calypogeia muelleriana* (Schiffn.) Müll.Frib.	*Calypogeia muelleriana* (Schiffn.) Müll.Frib.	Russia, Russian Far East, Magadan Province, V.A. Bakalin, Mag-31-26-12 (VBGI)	-	OL408984	-
*Calypogeia muelleriana* (Schiffn.) Müll.Frib.	*Calypogeia sphagnicola* f. *paludosa* (Warnst.) Schust.	Russia, Russian Far East, Magadan Province, V.A. Bakalin, Mag-32-41a-12 (VBGI)	-	OL408985	OL311455
*Calypogeia muelleriana* (Schiffn.) Müll.Frib.	*Calypogeia pseudointegristipula* Bakalin, Troizk. et Maltseva	Russia, Russian Far East, Sakhalin Province, Sakhalin Island, V.A. Bakalin, S-43-6-16 (VBGI)	-	OL408986	OL311457
*Calypogeia neesiana* (C.Massal. et Carestia) Müll.Frib.	*Calypogeia neesiana* (C.Massal. et Carestia) Müll.Frib.	Poland, POZW 41952	-	MK294001	-
*Calypogeia neesiana* (C.Massal. et Carestia) Müll.Frib.	*Calypogeia neesiana* (C.Massal. et Carestia) Müll.Frib.	Poland, K. Buczkowska, A. Baczkiewicz, POZW 41735	-	MH367820	* MH367694
*Calypogeia neesiana* (C.Massal. et Carestia) Müll.Frib.	*Calypogeia neesiana* (C.Massal. et Carestia) Müll.Frib.	Poland, K. Buczkowska, B. Chmielewski, POZW 41927	-	MH367821	* MH367695
*Calypogeia neesiana* (C.Massal. et Carestia) Müll.Frib.	*Calypogeia neesiana* (C.Massal. et Carestia) Müll.Frib.	Poland, K. Buczkowska, A. Baczkiewicz, POZW 41358	-	MH367822	* MH367696
*Calypogeia neesiana* (C.Massal. et Carestia) Müll.Frib.	*Calypogeia neesiana* (C.Massal. et Carestia) Müll.Frib.	Poland, POZW 41731	-	MK294002	-
*Calypogeia neesiana* (C.Massal. et Carestia) Müll.Frib.	*Calypogeia neesiana* (C.Massal. et Carestia) Müll.Frib.	Poland, K. Buczkowska, A. Baczkiewicz, POZW 41731	-	MH367819	MH367693
*Calypogeia neesiana* (C.Massal. et Carestia) Müll.Frib.	*Calypogeia neesiana* (C.Massal. et Carestia) Müll.Frib.	Russia, Russian Far East, Sakhalin Province, Sakhalin Island, S.V. Dudov, 2016_Br_0073	-	OL408987	OL311434
*Calypogeia neesiana* (C.Massal. et Carestia) Müll.Frib.	*Calypogeia muelleriana* (Schiffn.) Müll. Frib.	Russia, Russian Far East, Sakhalin Province, Sakhalin Island, S.V. Dudov & K.V. Kotelnikova, 2014_Br_0205	-	OL408988	OL409017
*Calypogeia neesiana* ssp. *subalpina* (Inoue) Inoue	*Calypogeia subalpina* Inoue	Japan, Honshu Island, V.A. Bakalin, J-87-1-15 (VBGI)	-	OL408993	OL311450
*Calypogeia neogaea* (R.M. Schust.) Bakalin	*Calypogeia kamchatica* Bakalin, Troizk. et Maltseva	Russia, Russian Far East, Kamchatka Territory, V.A. Bakalin, HRE, 28 (VBGI)	OL689060	OL408994	OL311445
*Calypogeia neogaea* (R.M. Schust.) Bakalin	*Calypogeia kamchatica* Bakalin, Troizk. et Maltseva	Russia, Russian Far East, Kamchatka Territory, V.A. Bakalin, K-63-1-15 (VBGI)	OL689061	OL408995	OL311446
*Calypogeia neogaea* (R.M. Schust.) Bakalin	*Calypogeia tosana* (Steph.) Steph.	Russia, Russian Far East, Sakhalin Province, Iturup Island, V.A. Bakalin, K-72-13-15 (VBGI)	OL689062	OL408996	OL311452
*Calypogeia neogaea* (R.M. Schust.) Bakalin	*Calypogeia neogaea* (R.M. Schust.) Bakalin	USA, Sainte Genevieve County, V.A. Bakalin, US-41-77-14 (VBGI)	OL689063	OL408997	OL311428
*Calypogeia orientalis* Buczkowska et Bakalin	*Calypogeia orientalis* Buczkowska et Bakalin	Russia, Russian Far East, Primorsky Territory, V.A. Bakalin, P-40-1-12 (VBGI)	-	* MH367790	MH367668
*Calypogeia orientalis* Buczkowska et Bakalin	*Calypogeia orientalis* Buczkowska et Bakalin	Japan, M. Higuchi, 02532867 (NYBG)	-	* MH367795	MH367673
*Calypogeia orientalis* Buczkowska et Bakalin	*Calypogeia orientalis* Buczkowska et Bakalin	Japan, K. Yamada, Yamada 8912	-	* MH367796	MH367674
*Calypogeia orientalis* Buczkowska et Bakalin	*Calypogeia orientalis* Buczkowska et Bakalin	Russia, Russian Far East, Primorsky Territory, V.A. Bakalin, P-15-12-12 (VBGI)	MH367553	MH367788	MH367666
*Calypogeia orientalis* Buczkowska et Bakalin	*Calypogeia orientalis* Buczkowska et Bakalin	Russia, Russian Far East, Primorsy Territory, V.A. Bakalin, P-39-4-12 (VBGI)	MH367554	MH367789	MH367667
*Calypogeia orientalis* Buczkowska et Bakalin	*Calypogeia orientalis* Buczkowska et Bakalin	Korea, Gangwon Province, V.A. Bakalin, Kor-7-23-11 (VBGI)	MH367555	* MH367791	* MH367669
*Calypogeia orientalis* Buczkowska et Bakalin	*Calypogeia orientalis* Buczkowska et Bakalin	Korea, Gangwon Province, V.A. Bakalin, Kor-10-02-11 (VBGI)	MH367556	* MH367792	MH367670
*Calypogeia orientalis* Buczkowska et Bakalin	*Calypogeia orientalis* Buczkowska et Bakalin	Korea, Gangwon Province, V.A. Bakalin, Kor-7-36-11 (VBGI)	MH367557	* MH367793	MH367671
*Calypogeia orientalis* Buczkowska et Bakalin	*Calypogeia orientalis* Buczkowska et Bakalin	Japan, Kyushu Island, V.A. Bakalin, J-7-79-14 (VBGI)	MH367558	* MH367794	MH367672
*Calypogeia peruviana* Nees et Mont.	*Calypogeia peruviana* Nees et Mont.	USA, North Carolina, B. Aguero, POZW 42619	MH367562	MH367800	MH367678
*Calypogeia peruviana* Nees et Mont.	*Calypogeia peruviana* Nees et Mont.	USA, North Carolina, B. Aguero, POZW 42627	MH367563	MH367801	* MH367679
*Calypogeia sinensis* Bakalin et Buczkowska	*Calypogeia sinensis* Bakalin et Buczkowska	Vietnam, Sơn La Province, V.A. Bakalin & K.G. Klimova, V-21-26-19 (VBGI)	-	-	OL311429
*Calypogeia sinensis* Bakalin et Buczkowska	*Calypogeia sinensis* Bakalin et Buczkowska	Vietnam, Sơn La Province, V.A. Bakalin & K.G. Klimova, V-23-15-19 (VBGI)	-	-	OL311430
*Calypogeia sinensis* Bakalin et Buczkowska	*Calypogeia sinensis* Bakalin et Buczkowska	China, Guizhou Proivince, V.A. Bakalin, China-56-77-13 (VBGI)	MH367559	MH367797	MH367675
*Calypogeia sinensis* Bakalin et Buczkowska	*Calypogeia sinensis* Bakalin et Buczkowska	China, Guizhou Proivince, V.A. Bakalin, China-56-78-13 (VBGI)	MH367560	* MH367798	MH367676
*Calypogeia sinensis* Bakalin et Buczkowska	*Calypogeia sinensis* Bakalin et Buczkowska	Vietnam, Lao Cai Province, V.A. Bakalin, V-2-73-16 (VBGI)	MH367561	* MH367799	MH367677
*Calypogeia sinensis* Bakalin et Buczkowska	*Calypogeia sinensis* Bakalin et Buczkowska	Vietnam, Hà Giang Province, V.A. Bakalin & K.G. Klimova, V-11-7-20 (VBGI)	OL689032	OL408962	OL311462
*Calypogeia sinensis* Bakalin et Buczkowska	*Calypogeia sinensis* Bakalin et Buczkowska	Vietnam, Cao Bằng Province, V.A. Bakalin & K.G. Klimova, V-27-4-20 (VBGI)	OL689035	OL408964	OL311469
*Calypogeia sinensis* Bakalin et Buczkowska	*Calypogeia sinensis* Bakalin et Buczkowska	Vietnam, Cao Bằng Province, V.A. Bakalin & K.G. Klimova, V-24-10-20 (VBGI)	OL689078	OL409011	OL311468
*Calypogeia sinensis* Bakalin et Buczkowska	*Calypogeia sinensis* Bakalin et Buczkowska	Vietnam, Hà Giang Province, V.A. Bakalin & K.G. Klimova, V-11-23-20 (VBGI)	OL689082	OL409014	OL311461
*Calypogeia* sp. nov.	*Calypogeia granulata* Inoue	Vietnam, Lao Cai Province, V.A. Bakalin, V-1-130-16 (VBGI)	OL689042	-	OL311447
*Calypogeia* sp.	*Calypogeia apiculata* (Steph.) Steph.	Vietnam, Lai Châu Province, V.A. Bakalin & K.G. Klimova, V-12-5-19 (VBGI)	OL689043	OL408969	OL311464
*Calypogeia* sp.	*Calypogeia pseudocuspidata* Bakalin, Frank Müll. et Troizk.	Vietnam, Lao Cai Province, V.A. Bakalin & K.G. Klimova, V-21-17-18 (VBGI)	OL689046	OL408972	OL311458
*Calypogeia* sp.	*Calypogeia* sp.	Russia, Russian Far East, Primorsky Territory, V.A. Bakalin, P-37-20-14	-	OL408973	-
*Calypogeia* sp.	*Calypogeia muelleriana* (Schiffn.) Müll.Frib.	Russia, Russian Far East, Khabarovsk Territory, K.G. Klimova, Khab-69-31-18 (VBGI)	OL689049	OL408977	-
*Calypogeia* sp.	*Calypogeia apiculata* (Steph.) Steph.	Vietnam, Lai Châu Province, V.A. Bakalin & K.G. Klimova, V-4-15-19 (VBGI)	OL689051	OL408979	OL311459
*Calypogeia* sp.	*Calypogeia* sp.	Japan, Honshu Island, M. Higuchi, BSE, 1293	OL689059	OL408992	-
*Calypogeia sphagnicola* (Arnell et J.Perss.) Warnst. et Loeske	*Calypogeia pseudosphagnicola* Bakalin, Troizk. et Maltseva	Russia, Russian Far East, Khabarovsk Territory, V.A. Bakalin, Kh-40-28-13 (VBGI)	-	-	OL311454
*Calypogeia sphagnicola* (Arnell et J.Perss.) Warnst. et Loeske	*Calypogeia sphagnicola* (Arnell et J.Perss.) Warnst. et Loeske	Russia, Russian Far East, Khabarovsk Territory, K.G. Klimova, Khab-68-14-18 (VBGI)	-	-	OL311432
*Calypogeia sphagnicola* (Arnell et J.Perss.) Warnst. et Loeske	*Calypogeia sphagnicola* (Arnell et J.Perss.) Warnst. et Loeske	Russia, Russian Far East, Magadan Province, V.A. Bakalin, Mag-38-16-12 (VBGI)	-	-	OL311433
*Calypogeia sphagnicola* (Arnell et J.Perss.) Warnst. et Loeske	*Calypogeia* sp.	Japan, H. Inoue, KRAM 50656	-	-	MH367709
*Calypogeia sphagnicola* (Arnell et J.Perss.) Warnst. et Loeske	*Calypogeia sphagnicola* (Arnell et J.Perss.) Warnst. et Loeske	Russia, Russian Far East, Kamchatka Territory, V.A. Bakalin, K-54-1-15 (VBGI)	-	OL408998	OL311431
*Calypogeia sphagnicola* (Arnell et J.Perss.) Warnst. et Loeske	*Calypogeia pseudosphagnicola* Bakalin, Troizk. et Maltseva	Russia, Russian Far East, Sakhalin Province, Sakhalin Island, V.A. Bakalin, S-17-3-17 (VBGI)	-	OL408999	OL311456
*Calypogeia sphagnicola* (Arnell et J.Perss.) Warnst. et Loeske f. *sphagnicola*	*Calypogeia sphagnicola* (Arnell et J.Perss.) Warnst. et Loeske f. *sphagnicola*	NW Poland, K. Buczkowska & A. Bączkiewicz, POZW 42284	-	JQ658790	JQ658773
*Calypogeia sphagnicola* (Arnell et J.Perss.) Warnst. et Loeske f. *sphagnicola*	*Calypogeia sphagnicola* (Arnell et J.Perss.) Warnst. et Loeske f. *sphagnicola*	NW Poland, K. Buczkowska & A. Bączkiewicz, POZW 42245	-	JQ658791	* JQ658774
*Calypogeia sphagnicola* (Arnell et J.Perss.) Warnst. et Loeske f. *sphagnicola*	*Calypogeia sphagnicola* (Arnell et J.Perss.) Warnst. et Loeske f. *sphagnicola*	Poland, K. Buczkowska & A. Bączkiewicz, POZW 42351	-	JQ658792	* JQ658775
*Calypogeia sphagnicola* (Arnell et J.Perss.) Warnst. et Loeske f. *sphagnicola*	*Calypogeia sphagnicola* (Arnell et J.Perss.) Warnst. et Loeske f. *sphagnicola*	NW Poland, K. Buczkowska & A. Bączkiewicz, POZW 42266	-	JQ658793	* JQ658776
*Calypogeia sphagnicola* (Arnell et J.Perss.) Warnst. et Loeske f. *sphagnicola*	*Calypogeia sphagnicola* (Arnell et J.Perss.) Warnst. et Loeske f. *sphagnicola*	NE Poland, K. Buczkowska & A. Bączkiewicz, POZW 41711	-	JQ658794	JQ658777
*Calypogeia sphagnicola* f. *paludosa* (Warnst.) Schust.	*Calypogeia sphagnicola* f. *paludosa* (Warnst.) Schust.	S Poland, K. Buczkowska & A. Bączkiewicz, POZW 41174	-	JQ658795	JQ658778
*Calypogeia sphagnicola* f. *paludosa* (Warnst.) Schust.	*Calypogeia sphagnicola* f. *paludosa* (Warnst.) Schust.	S Poland, K. Buczkowska & A. Bączkiewicz, POZW 41148	-	JQ658796	JQ658779
*Calypogeia sphagnicola* f. *paludosa* (Warnst.) Schust.	*Calypogeia sphagnicola* f. *paludosa* (Warnst.) Schust.	S Poland, K. Buczkowska & A. Bączkiewicz, POZW 42277	-	JQ658797	* JQ658780
*Calypogeia sphagnicola* f. *paludosa* (Warnst.) Schust.	*Calypogeia sphagnicola* f. *paludosa* (Warnst.) Schust.	S Poland, K. Buczkowska & A. Bączkiewicz, POZW 41722	-	JQ658798	* JQ658781
*Calypogeia suecica* (Arnell et J.Perss.) Müll. Frib.	*Calypogeia suecica* (Arnell et J.Perss.) Müll. Frib.	Poland, J. Szweykowski, K. Buczkowska, POZW 39500	-	* MH367815	MH367689
*Calypogeia suecica* (Arnell et J.Perss.) Müll. Frib.	*Calypogeia suecica* (Arnell et J.Perss.) Müll. Frib.	Russia, Russian Far East, Khabarovsk Territory, K.G. Klimova, Khab-60-34-18 (VBGI)	-	-	OL311436
*Calypogeia suecica* (Arnell et J.Perss.) Müll. Frib.	*Calypogeia* sp.	Russia, Russian Far East, Primorsky Territory, V.A. Bakalin & G. Arutinov, P-2-6a-13 (VBGI)	-	-	OL311460
*Calypogeia suecica* (Arnell et J.Perss.) Müll. Frib.	*Calypogeia suecica* (Arnell et J.Perss.) Müll. Frib.	Russia, Russian Far East, Primorsky Territory, V.A. Bakalin, P-37-1-14 (VBGI)	-	-	OL311437
*Calypogeia sullivantii* Austin	*Asperifolia sullivantii* (Austin) Troizk., Bakalin, Maltseva	USA, North Carolina, B. Aguero, POZW 42623	MH367569	MH367813	MH367687
*Calypogeia sullivantii* Austin	*Asperifolia sullivantii* (Austin) Troizk., Bakalin, Maltseva	USA, Maryland, L.T. Biechele, NYGB 2169286	MH367570	MH367814	* MH367688
*Calypogeia tosana* (Steph.) Steph.	*Calypogeia tosana* (Steph.) Steph.	China, Guizhou Province, V.A. Bakalin, C-55-72-13 (VBGI)	OL689067	-	OL311438
*Calypogeia tosana* (Steph.) Steph.	*Calypogeia tosana* (Steph.) Steph.	China, Guizhou Proivince, V.A. Bakalin, China-56-86-13 (VBGI)	-	MH367807	* MH367683
*Calypogeia tosana* (Steph.) Steph.	*Calypogeia tosana* (Steph.) Steph.	Japan, Kyushu Island, Fukuoka Prefecture, V.A. Bakalin, J-4-38-14 (VBGI)	-	MH367808	MH367684
*Calypogeia tosana* (Steph.) Steph.	*Calypogeia tosana* (Steph.) Steph.	Vietnam, Lao Cai Province, V.A. Bakalin, V-1-102-16 (VBGI)	OL689033	OL408963	-
*Calypogeia tosana* (Steph.) Steph.	*Calypogeia tosana* (Steph.) Steph.	Vietnam, Cao Bằng Province, V.A. Bakalin & K.G. Klimova, V-23-2-20 (VBGI)	OL689040	OL408966	OL311471
*Calypogeia tosana* (Steph.) Steph.	*Calypogeia tosana* (Steph.) Steph.	China, Guizhou Province, V. A. Bakalin, C-55-19-13 (VBGI)	-	OL408978	OL311479
*Calypogeia tosana* (Steph.) Steph.	*Calypogeia tosana* (Steph.) Steph.	Vietnam, Lai Châu Province, V.A. Bakalin & K.G. Klimova, V-16-2-19 (VBGI)	OL689052	OL408980	OL311440
*Calypogeia tosana* (Steph.) Steph.	*Calypogeia tosana* (Steph.) Steph.	Republic of Korea, Jeollabuk-do, V.A. Bakalin & S.S. Choi, Kor-73-16-19 (VBGI)	OL689058	OL408991	OL311435
*Calypogeia tosana* (Steph.) Steph.	*Calypogeia tosana* (Steph.) Steph.	Republic of Korea, Jeollabuk-do, V.A. Bakalin & S.S. Choi, Kor-79-25-19 (VBGI)	OL689066	OL409002	OL311439
*Calypogeia tosana* (Steph.) Steph.	*Calypogeia yoshinagana* Steph.	Japan, Kyushu Island, V.A. Bakalin, J-40-52-14 (VBGI)	OL689068	OL409003	OL311449
*Calypogeia tosana* (Steph.) Steph.	*Calypogeia tosana* (Steph.) Steph.	Russia, Russian Far East, Sakhalin Province, Iturup Island, V.A. Bakalin, K-70-35-15 (VBGI)	OL689069	OL409004	OL311451
*Calypogeia tosana* (Steph.) Steph.	*Calypogeia yoshinagana* Steph.	Republic Of Korea, Jeju Island, V.A. Bakalin, Kor-29-64-15 (VBGI)	OL689070	OL409005	-
*Calypogeia tosana* (Steph.) Steph.	*Calypogeia tosana* (Steph.) Steph.	Japan, Honshu Island, V.A. Bakalin, J-5-9-13 (VBGI)	-	OL409009	-
*Calypogeia vietnamica* Bakalin et Vilnet	*Calypogeia vietnamica* Bakalin et Vilnet	Vietnam, Lao Cai Province, V.A. Bakalin & K.G. Klimova, V-8-61-17 (VBGI)	MK335824	MK336253	-
*Calypogeia vietnamica* Bakalin et Vilnet	*Calypogeia vietnamica* Bakalin et Vilnet	Vietnam, Lao Cai Province, V.A. Bakalin & K.G. Klimova, V-9-23-17 (VBGI)	MK335825	MK336254	-
*Calypogeia yoshinagana* Steph.	*Calypogeia yoshinagana* Steph.	Republic of Korea, Jeollabuk-do, V.A. Bakalin & S.S. Choi, Kor-72-5-19 (VBGI)	OL689075	-	OL311442
*Calypogeia yoshinagana* Steph.	*Calypogeia yoshinagana* Steph.	Republic of Korea, Jeollabuk-do, V.A. Bakalin & S.S. Choi, Kor-79-22-19 (VBGI)	OL689077	-	OL311443
*Calypogeia yoshinagana* Steph.	*Calypogeia yoshinagana* Steph.	Republic of Korea, Jeollabuk-do, V.A. Bakalin & S.S. Choi, Kor-82-3-19 (VBGI)	OL689081	-	OL311444
*Calypogeia yoshinagana* Steph.	*Calypogeia yoshinagana* Steph.	Republic of Korea, Gyeongsangnam-do, V.A. Bakalin & S.S. Choi, Kor-70-1-19 (VBGI)	OL689073	OL409007	OL311441
*Geocalyx graveolens* (Schrad.) Nees	*Geocalyx graveolens* (Schrad.) Nees	China, Yunnan Province, D.G. Long, 34828 (E)	-	* KJ802067	KJ802038
*Mesoptychia sahlbergii* (Lindb. et Arnell) A. Evans	*Mesoptychia sahlbergii* (Lindb. et Arnell) A. Evans	USA, D.G. Long, 11329 (E)	KJ206417	-	-
*Mesoptychia gillmanii* (Austin) L. Söderstr. et Váňa	*Mesoptychia gillmanii* (Austin) L. Söderstr. et Váňa	Sweden, A. Cailliau & N. Lonnell, 982 (G)	KJ206404	* KJ206363	-
*Mesoptychia gillmanii* (Austin) L. Söderstr. et Váňa	*Mesoptychia gillmanii* (Austin) L. Söderstr. et Váňa	Russia, N.A. Konstantinova & A.N. Savchenko, 117-1-00 (F)	-	* KJ802075	KJ802047
*Mesoptychia heterocolpos* (Thed. ex Hartm.) L. Söderstr. et Váňa	*Mesoptychia heterocolpos* (Thed. ex Hartm.) L. Söderstr. et Váňa	USA, Alaska, B. Shaw, F965a/3 (DUKE)	-	* KF943057	KF942896
*Mesoptychia heterocolpos* (Thed. ex Hartm.) L. Söderstr. et Váňa	*Mesoptychia heterocolpos* (Thed. ex Hartm.) L. Söderstr. et Váňa	Sweden, A. Cailliau, 999 (G)	KJ206411	* KJ206371	-
*Mesoptychia heterocolpos* (Thed. ex Hartm.) L. Söderstr. et Váňa	*Mesoptychia heterocolpos* (Thed. ex Hartm.) L. Söderstr. et Váňa	Russia, Komi Rep., Dulin M.V., 116749 (KPABG)	-	* KM501496	KM501477
*Mesoptychia heterocolpos* var. *harpanthoides* (Bryhn et Kaal.) L. Söderstr. et Váňa	*Mesoptychia heterocolpos* var. *harpanthoides* (Bryhn et Kaal.) L. Söderstr. et Váňa	Russia, Arkhangelsk Prov., Franz Josef Land, Ziegler Island, A. Savchenko, CA-19-29-8c (KPABG)	-	* MT431408	MT431398
*Mesoptychia sahlbergii* (Lindb. et Arnell) A. Evans	*Mesoptychia sahlbergii* (Lindb. et Arnell) A. Evans	Russia, V. Fedosov, 107967 (F)	-	* KJ802078	KJ802050
*Metacalypogeia alternifolia* (Nees) Grolle	*Metacalypogeia alternifolia* (Nees) Grolle	Bhutan, D.G. Long, 28712 (E)	-	KJ802068	KJ802040
*Metacalypogeia cordifolia* (Steph.) Inoue	*Metacalypogeia cordifolia* (Steph.) Inoue	Russia, Russian Far East, Primorsky Territory, V.A. Bakalin, P-66-18a-06 (VBGI)	* JX629934	JF421597	-
*Mnioloma fuscum* (Lehm.) R.M. Schust.	*Mnioloma fuscum* (Lehm.) R.M. Schust.	South Africa, de Roo, s.n. (BOL)	-	-	AM397718
*Mnioloma fuscum* (Lehm.) R.M. Schust.	*Mnioloma fuscum* (Lehm.) R.M. Schust.	Fiji, J.E. Braggins et al. 16 Apr 2008 (NSW)	-	KJ802100	-

* GenBank numbers marked with an asterisk are not included in the phylogenetic trees, because of their supposed heterogeneity.

**Table 2 plants-11-00983-t002:** Intergeneric *p*-distances over sequence pairs between groups. The number of base differences per site from averaging over all sequence pairs between groups is shown.

Taxon	Intergeneric *p*-Distances, *trn*G-Intron/*trn*L−F/ITS2				
	*Calypogeia*	*Asperifolia*	*Mnioloma*	*Metacalypogeia*	*Geocalyx*
*Asperifolia*	0.17/0.13/0.13				
*Mnioloma*	0.17/0.16/-	0.17/0.16/-			
*Metacalypogeia*	0.15/0.12/-	0.21/0.15/-	0.20/0.16/-		
*Geocalyx*	0.15/0.17/-	0.2/0.18/-	0.19/0.18/-	0.18/0.14/-	
*Mesoptychia*	0.16/0.15/0.19	0.19/0.17/0.22	0.18/0.19/-	0.19/0.13/-	0.16/0.13/-

**Table 3 plants-11-00983-t003:** Primers used in the polymerase chain reaction (PCR) and cycle sequencing reactions.

Locus	Sequence (5′-3′)	Direction	Annealing Temperature (°C)	Reference
ITS2 nrDNA	GCATCGATGAAGAACGCAGC	forward	62	[33]
ITS2 nrDNA	GATATGCTTAAACTCAGCGG	reverse	58	[33]
ITS2 nrDNA	GAGTCATCAGCTCGCGTTGAC	forward	66	[34]
ITS2 nrDNA	GCTGCGTTCTTCATCGATGC	reverse	62	[35]
*trn*L–F cpDNA	CGAATTCGGTAGACGCTACG	forward	62	[36]
*trn*L–F cpDNA	ATTTGAACTGGTGACACGAG	reverse	58	[36]
*trn*L–F cpDNA	CGAAATTGGTAGACGCTGCG	forward	62	[37]
*trn*L–F cpDNA	TGCCAGAAACCAGATTTGAAC	reverse	60	[37]
*trn*G-intron cpDNA	ACCCGCATCGTTAGCTTG	forward	56	[38]
*trn*G-intron cpDNA	GCGGGTATAGTTTAGTGG	reverse	54	[38]

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
