# Peer review of "Calypogeia (Calypogeiaceae, Marchantiophyta) in Pacific Asia: Updates from Molecular Revision with Particular Attention to the Genus in North Indochina"

_plants, 2022, doi:10.3390/plants11070983_

Round 1

Reviewer 1 Report

I'm impressed with your extensive studies and detailed report.  The primary area requiring further work would be to polish the English to be clearer and more succinct.  To that end, I invested considerable time editing your manuscript.  I have attached a marked-up version with all suggested changes highlighted in yellow and also including some "sticky notes" with items needing special attention.

Author Response

Thank you very much for your very exhaustive reading of the manuscript that made it much better.

Below are our answers:

1) The English was improved with particular attention to the sites you indicated; although sometimes we could not understand what changes in language usage should be applied.

2) We explained scale bars measurements as you required for phylogenetic tree figures.

3) We changed the colors in the Figure 3 as you requested and added note on the nature of the gaps (yes, the left bands are infra-generic, the right bands are inter-generic)

4) We provided the necessary reference to azulene derivates research as you requested (and changed reference numeration accordingly).

5) The reference list was re-checked, improved, the requested information was added.

Reviewer 2 Report

The manuscript by Bakalin et al. presents interesting, original research. This is a very well-formulated paper that has clearly taken a significant amount of research, including extensive field and lab work. This is a very relevant study, but there are several points throughout the manuscript which need to be addressed before publication.

The main issue I see is presentation of results of phylogenetic analysis and the analysis itself. In addition to nuclear ITS2 sequences, the authors used two chloroplast markers - trnG intron and trnL-F region - and performed their separate phylogenetic analysis. As plastid genome sequences are expected to share a common phylogeny, it is better to combine them in a simultaneous phylogenetic analysis, this approach helps to reduce stochastic error in phylogenetic inference. The second point is the authors presented fully resolved Bayesian trees, while many internal branches gained very week support in different analyses and should be collapsed. Indeed, short plastid sequences will not resolve some relationships within Calypogeia, the tree will be mostly unresolved in backbone, and such a picture would be more objective. To do this, the authors should use a command "sumt" with the option "halfcompat" instead of "allcompat" in MrBayes, the same concerns the ITS2 analysis.

Minor comments:

Lines 74, 109, and 120: for many years the term "autapomorphic character" has been used in phylogenetics, why use "singleton"?

Line 97: remove "per site", as p-distance itself shows a number of substitutions per site

Table 2: "intergeneric" in the table legend and "infrageneric" in the Table 2, correct this and remove a percent sign (see above)

Lines 227-228: "...is not genetically related...", but generally all living things are related, just some ones are more closely related than others

Author Response

Reviewer 2

1) Lines 74, 109, and 120: for many years the term "autapomorphic character" has been used in phylogenetics, why use "singleton"?

Agreed, improved

2) Line 97: remove "per site", as p-distance itself shows a number of substitutions per site

Agreed, improved

3) Table 2: "intergeneric" in the table legend and "infrageneric" in the Table 2, correct this and remove a percent sign (see above)

Thank you very much (this is our mistake), we have improved, the correct meaning is intergeneric

4) Lines 227-228: "...is not genetically related...", but generally all living things are related, just some ones are more closely related than others

Agreed, changed for “is not closely genetically related”

5) As plastid genome sequences are expected to share a common phylogeny, it is better to combine them in a simultaneous phylogenetic analysis, this approach helps to reduce stochastic error in phylogenetic inference.

Not agreed. The phylogenetic trees based on trnG and trnL-F are similar in general traits (e.g. the distance of Asperifolia from Calypogeia s.l.) but different in many details. Formally, this is not preventing to make the combined tree, although this may affect the result. We avoid the providing the combined phylogenetic tree because several specimens should be eliminated from the analysis: we have only trnG or trnL-F for a couple of specimens.

6) The second point is the authors presented fully resolved Bayesian trees, while many internal branches gained very week support in different analyses and should be collapsed. Indeed, short plastid sequences will not resolve some relationships within Calypogeia, the tree will be mostly unresolved in backbone, and such a picture would be more objective. To do this, the authors should use a command "sumt" with the option "halfcompat" instead of "allcompat" in MrBayes, the same concerns the ITS2 analysis.

Agreed, the condensed trees are provided n supplementary materials.